# A study of autoencoders as a feature extraction technique for spike sorting

**Eugen-Richard Ardelean** [1,2] *, **Andreea Coporîie** [2], **Ana-Maria Ichim** [1], **Mihaela Dînşoreanu** [2], **Raul Cristian Mureşan** [1,3]

**1** Department of Experimental and Theoretical Neuroscience, Transylvanian Institute of Neuroscience, Cluj-Napoca, Romania, **2** Department of Computer Science, Technical University of Cluj-Napoca, Cluj-Napoca, Romania, **3** STAR-UBB Institute, Babeş-Bolyai University, Cluj-Napoca, Romania

* ardeleaneugenrichard@gmail.com

## Abstract

Spike sorting is the process of grouping spikes of distinct neurons into their respective clusters. Most frequently, this grouping is performed by relying on the similarity of features extracted from spike shapes. In spite of recent developments, current methods have yet to achieve satisfactory performance and many investigators favour sorting manually, even though it is an intensive undertaking that requires prolonged allotments of time. To automate the process, a diverse array of machine learning techniques has been applied. The performance of these techniques depends however critically on the feature extraction step. Here, we propose deep learning using autoencoders as a feature extraction method and evaluate extensively the performance of multiple designs. The models presented are evaluated on publicly available synthetic and real "in vivo" datasets, with various numbers of clusters. The proposed methods indicate a higher performance for the process of spike sorting when compared to other state-of-the-art techniques.

## 1. Introduction

### 1.1. Spike sorting

The individual activation events of a neuron are called action potentials or spikes. Attributing such a spike to the specific neuron that produced it based on its characteristics is referred to as spike sorting [1]. Spike sorting, by definition, handles extracellular recordings, which capture the activity of multiple neurons in the proximity of the recording electrode [2] thus, the neuron of a spike is unknown at the time of recording. The main assumption of spike sorting is that each distinct neuron tends to generate spikes of similar shapes [3], yet markedly different from the shapes of spikes of other neurons. In reality, the shape of spikes is muddled by the background noise, inducing variability, which generates a cluster in the feature space instead of a single point. Therefore, it is important to find or generate features that are able to separate the spikes and that are preferably as few as possible.

The spike sorting pipeline can be broken up into four sequential steps [1]: filtering, spike detection, feature extraction, and clustering. Importantly, the separability of clusters is driven by the feature extraction technique and not by the clustering method. Here, we investigate the

**Data Availability Statement:** Code Availability: The code that supports the findings of this study was written in Python and is openly available at https://github.com/ArdeleanRichard/SpikeSorting. Data Availability: The synthetic data used in this work is

openly accessible at https://doi.org/10.1016/j.
jneumeth.2012.07.010, and it can also be found at
https://1drv.ms/u/s!
AgNd2yQs3Ad0gSjeHumstkCYNcAk?e=QfGIJO or
https://www.kaggle.com/datasets/ardeleanrichard/
simulationsdataset. The real mouse data can be
found in the git repository along with the code:
https://github.com/ArdeleanRichard/Autoencoders-
in-Spike-Sorting or https://www.kaggle.com/
datasets/ardeleanrichard/realdata.

**Funding:** The research leading to these results has
received funding from: NO (Norway) Grants 2014-
2021, under Project contract number 20/2020
(RO-NO-2019-0504), four grants from the Ro-
manian National Authority for Scientific Research
and Innovation, CNCS-UEFISCDI (codes PN-III-P2-
2.1-PED-2019-0277, PN-III-P3-3.6-H2020-2020-
0109, ERA-NET-FLAG-ERA-
ModelDXConsciousness, and ERANET-NEURON-
Unscrambly), and a H2020 grant funded by the
European Commission (grant agreement 952096,
NEUROTWIN). The funders had no role in study
design, data collection and analysis, decision to
publish, or preparation of the manuscript.

**Competing interests:** The authors have declared
that no competing interests exist.

impact of different feature extraction techniques on the separability of resulting clusters, nevertheless the other steps of the spike sorting are topics of ongoing research in this domain as well. We would like to draw attention to the fact that a golden feature extraction method does not exist and the performance of each depends on the characteristics of the data [1, 4]. Here, we employ autoencoders in an unsupervised learning paradigm to discover relevant features and compare its impact on the performance of spike-sorting pipelines with state-of-the-art methods that use other feature spaces.

The spike sorting pipeline can be modified depending on the approach used, offline or online. In offline spike sorting, the sorting is done only after the data acquisition, while in online it happens during. In the filtering step of the raw signal, a band-pass filter is applied in order to isolate the relevant frequency band (usually between 300 and 3000Hz [4]) where the spike's frequency components are expressed. Next, spike detection typically involves amplitude thresholding, while compromising between missing spikes and including noise in the data. The third step, and the focus of this study, is the feature extraction step, whereby the most informative features are identified and extracted in order to reduce the dimensionality of the data and reduce the computation load of the clustering while maintaining the data separability. In the final steps, spikes are clustered in the feature space such that similar spikes are separated into groups, each group assumed to have been generated by the same neuron. Alternatively, a supervised manual approach was commonly used where the researcher could classify spikes by hand. Nonetheless, such methods are rapidly becoming impractical as new multi-array probes are developed [5] as the number of recorded neurons has seen an exponential increase since the 1950s [6]. A template matching approach that is applied on a subsampled set of data has become increasingly popular and can substitute of the spike detection and feature extraction steps, while also being computationally efficient [7].

## 1.2. Autoencoders

Autoencoders (AE) are neural networks that have the ability to learn the underlying features of unlabelled data [8, 9]. Autoencoders are most commonly consisting of two linked parts: the encoder and the decoder, as shown in Fig 1A. The encoder maps the input into a latent representation or code, while the decoder attempts to make a reconstruction of the input from the intermediate code. Over time, many variations of autoencoders have been developed: sparse, denoising, contractive, and many others [8]. Each of these variations is optimized for certain tasks. Autoencoders have found applications in a diverse array of domains: generative modelling, anomaly detection, recommendation systems, feature extraction for classification and clustering, and dimensionality reduction [8].

The usefulness of Autoencoders as a dimensionality reduction technique has been demonstrated on image datasets [10–12], such as MNIST. Similar to other feature extraction methods,

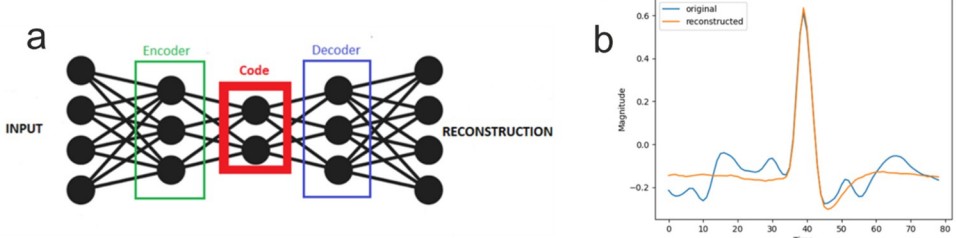

**Fig 1. The Autoencoder. (a)** Diagram of the autoencoder components **(b)** Original spike and the reconstruction made by an autoencoder.

autoencoders discover a low-dimensional pattern of variation in the data that retains enough information to restore the input [8]. Thus, the quality of an autoencoder is given by its ability to restore the input. An example of spike reconstruction using autoencoders is shown in Fig 1B.

The latent code's succinct representation contains relevant information about the original input and it constitutes a feature extraction method. Being neural networks, the encoder and decoder allow for custom architectures. Thus, increasing the complexity of the architecture will result in the generation of more intricate latent codes.

### 1.3. The challenges of spike sorting

The process of spike sorting is challenging due to an array of difficulties. First, because neuronal firing occurs on millisecond timescales, even relatively brief recordings generate an abundant data volume [3]. Second, rather than being stationary, the activity of neurons is regulated by brain circuits such that they can fire with markedly different firing rates [13, 14]. This results in different relative frequencies at different times, leading to clusters of different sizes and an inherent imbalance in the data. Many clustering algorithms have difficulties tackling imbalanced data especially when coupled with overlap. Finally, in practice various phenomena can alter or contaminate the estimated spike shape, such that clusters are not always distinct, but often overlap. Single unit activity is defined as the activity of a single neuron that can be separated as a single cluster, while the activity of distal neurons is represented in the signal as low amplitude spikes and most often cannot be separated due to a low signal-to-noise ratio and as such, is denominated as multiunit activity [4].

The aim is to create a representation that is unaffected by slight changes in waveform shape as a result of noise and phenomena such as the electrode drift that may modify the shape of the waveform. The low dimensional representation of autoencoders has been used on various types of data. The pattern of encoding offered by autoencoders has been proposed to be invariant to noise as they have been demonstrated to be a well-rounded denoising technique [8] providing robustness. Recently, autoencoders have received some attention in spike sorting [15, 16]. Nevertheless, these studies are limited to synthetic data with a relatively low number of clusters. In [15], an ensemble of autoencoders of different depths are used to extract the features of spikes, the proposed approach had a high accuracy when compared with conventional methods but the analysed datasets were restricted to a maximum of 5 different source neurons, each of them generating a single cluster of spikes. The evaluation of the extracted features was made using a single metric, the Silhouette [17, 18] score. In [16], the authors propose contractive autoencoders for the problem of spike sorting. The evaluation of performance used accuracy as a metric, which tends to incorrectly assess the performance for imbalanced classes and is only applicable for synthetic datasets the provide a ground truth. Because spike sorting handles inherently imbalanced datasets, more specialized metrics are required to evaluate performance.

The paper is organized as follows: section 2 presents a critical view of conventional feature extraction methods used in spike sorting, provides a description of the proposed method, and presents the datasets and metrics used in the analysis. In section 3, the methods are evaluated considering multiple metrics and their performances are interpreted critically. Section 4 discusses the limits of the proposed method and the conclusions we have reached.

## 2. Materials and methods

### 2.1. State of the art

As stated above, a crucial step in spike sorting is the description of spikes with a compact set of informative features. The aim of dimensionality reduction is to transform a dataset with a

dimensionality of X into a dataset with Y dimensions, where Y<<X. Another important aim is to retain as much of the data geometry as possible, such that relations in the original space are retained in the reduced space, which is especially useful for spike sorting. Dimensionality reduction techniques can be divided by several criteria, such as: convexity or linearity [19]. From the point of view of convexity, PCA is a convex algorithm, while autoencoders are a non-convex approach. Among the first features used in the spike sorting were the spike amplitude and its width [20]. Afterward, methods based on probabilistic models, created through empirical analysis, that used the entire waveform were developed [21]. These could process a low number of electrodes. Shortly thereafter, transforms started being used to project the high-dimensional space of the waveform into a low-dimensional space through the use of principal components [22], the wavelet transform [23] and various combinations of them. Manual sorting of spikes is usually performed on a low dimensional space, containing features such as the amplitude, the peak-to-trough ratio, etc [24]. The peak-to-trough ratio was found to be representative of the neuron type, inhibitory neurons produce narrow spikes and thus have a small peak-to-trough ratio, while excitatory have a large ratio [25].

In [26], the authors propose M-Sorter, an automatic method for spike detection and classification based on coefficients obtained through the wavelet transform and template matching. The proposed method separates spike sorting into two steps, the spike detection by multiple correlation of wavelet coefficients on the band pass filtered waveforms of the recorded signal and template matching for the classification of spikes to the neurons that generated it. The multiple correlation of wavelet coefficients is also used in the generation of templates through the application of K-Means. Each spike is assigned to the cluster to which it has the smallest distance.

**2.1.1. Linear feature extraction methods.** Principal Component Analysis (PCA) [27] is the most frequently used algorithm for feature extraction, including spike sorting [28]. PCA projects the spikes onto new characteristics called Principal Components that are a new set of orthogonal axes formed by linear combinations of the input features. The reduction of dimensionality of the feature space is performed by solving a problem of eigenvalues and eigenvectors. By retaining the most prominent principal components, PCA preserves the variance as much as possible while being able to reduce the number of features. It is common to keep only the first two or three principal components resulted from PCA [29, 30]. These frequently retain more than 70% of the variance from the original space. However, variance does not necessarily offer the best separation [1, 4]. To put it in another way, information required for separability may be encoded in those low-variance features that are discarded. Finally, PCA and its variations have been used in spike sorting for a long time [4] and it is still used in recently developed spike sorting pipelines [31].

Another linear method is Independent Component Analysis (ICA) [32] mainly designed for source separation. ICA is a linear unsupervised technique for dimensionality reduction that searches for independent components by relying on the statistical properties of the data. ICA has been previously applied to spike sorting with promising results [33, 34].

Linear Discriminant Analysis (LDA) [35] is a supervised linear learning technique with the goal of increasing the inter-cluster distance and decreasing intra-cluster distance. LDA assumes that the data has a Gaussian distribution. However, for our problem LDA is not a fit candidate due to several considerations. First, it is a supervised learning technique which cannot be applied to unlabelled data, as is the case in spike sorting. Second, the Gaussian distribution assumption is often violated in spike sorting due to: electrode drift, shape variation from bursts, simultaneous firing, multi-unit activity, and non-stationary background noise [1].

**2.1.2. Non-linear feature extraction methods.** In the category of unsupervised non-linear dimensionally reduction techniques Isomap [36] uses Isometric Mapping to learn the low-

dimensional projection in a manifold space while retaining the distances of the original space. It uses the geodesic distance, which can be thought of as the shortest path along the curved surface of the manifold space.

T-distributed Stochastic Neighbor Embedding (t-SNE) [37] is a non-linear dimensionality reduction method that minimizes the divergence between input features and the reduced feature space by using pairwise probability similarities. The divergence of two distributions is calculated using KL divergence, which is minimized by applying gradient descent. Due to its high time complexity, several orders of magnitude higher than PCA, and its main function being visualization, t-SNE was not considered a suitable candidate. A computation of a few seconds for PCA can become tens of minutes for t-SNE. Furthermore, from empirical observations, the separation offered by t-SNE for the datasets used here was small to non-existent.

## 2.2. Model architecture

In this section, we present the autoencoder variants that can be framed as a non-linear feature extraction method. In most cases, these architectures contain multiple hidden layers as they have been shown to better represent the details of the signal [15], and, consequently, could be branded as 'deep networks'. The encoder and decoder have mirrored, symmetric architectures, with *ReLU* activation functions, while the code and output layers use the *tanh* activation function. For the code layer and the output layer, the *tanh* activation function has been chosen instead of *ReLU* because it allows for negative values, which are needed in spike reconstruction and due to its nonlinearity. In addition, the code layer contains L1 regularization of 10e-7 to prevent overfitting. These architectures use Adam optimization with a learning rate of 0.001 and the Mean Squared Error (MSE) as a loss function. As previously mentioned, the reconstruction capability of the autoencoder is correlated to its performance, thus MSE is a suitable loss function because it will provide the best point-by-point approximation of the overall spike. In section III, we also present a short evaluation of the impact of parameters, such as learning rate or number of epochs, on performance. The evaluated Autoencoder (AE) variants follow the structure shown in Fig 2.

Next, we will present the Autoencoder (AE) variants in a list with a short description. Unless stated otherwise, the variants respect the specifications presented above. Details will be presented afterward in the paragraphs succeeding the list:

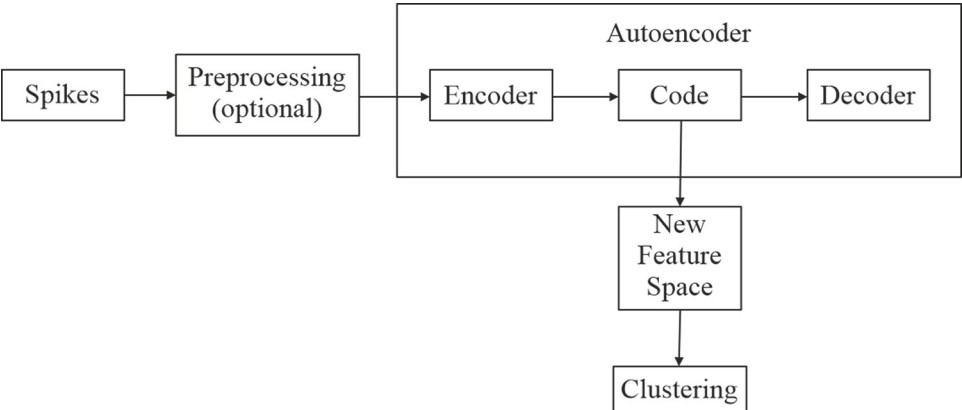

**Fig 2. The general model of autoencoder for feature extraction.** The input spikes are used to train the autoencoder. Once trained, the values obtained at the code layer can be extracted as a new feature space that is used in the clustering of the spikes. The preprocessing of the input data before training is optional.

1. **Shallow AE**: A shallow autoencoder with 3 fully connected layers on each subpart, the number of neurons per layer being decremented (60, 40, 20) in the encoder until the code of size 2 is reached, where the symmetric decoder contains incrementally larger layers.

2. **AE**: A deep autoencoder with 8 fully connected layers on each subpart, the number of neurons per layer being decremented (70,60,50,40,30,20,10,5) in the encoder until the code of size 2 is reached, where the symmetric decoder contains incrementally larger layers.

3. **Tied AE**: This autoencoder variant follows the structure and the specification of AE with the corresponding (by size) layers of the encoder and decoder tied [38]. The weight-sharing provided by this variant is able to reach convergence faster and it has been shown to have more accurate reconstructions [38] when compared to untied variants.

4. **PCA AE**: A deep autoencoder with 5 fully connected layers on each subpart, the number of neurons per layer being decremented (70,60,50,40,30) in the encoder until the code of size 20 is reached, where the symmetric decoder contains incrementally larger layers. The code provided by the training is then processed through PCA. As such, it is a combination of PCA and an autoencoder.

5. **Pretrained AE**: This autoencoder variant respects the specification of AE with the addition of a greedy layer-wise pretraining [39]. Pretraining provides an initialization of the weights of the autoencoder and has been shown to increase performance [39].

6. **LSTM AE**: A shallow autoencoder with 3 fully connected LSTM [40] layers. It respects the incrementing-decrementing format of the previous variants with a code size of 2.

7. **FT AE**: This autoencoder variant respects the specification of AE with an added preprocessing step consisting of applying the Fourier Transform [41] on the input data.

8. **WFT AE**: This autoencoder variant is similar to FT AE but the Fourier Transform is applied using a Blackman Window (W) in order to amplify the information of the amplitude that was aligned to the middle of the window.

9. **Orthogonal AE**: This autoencoder variant respects the specification of AE modified to ensure the orthogonality of the weights and the encoding of uncorrelated features. This variant has been shown to outperform PCA when combined with clustering [42].

10. **Contractive AE**: This autoencoder variant respects the specification of AE with a modified loss function using a regularization term that provides more robustness to noise [43].

**2.2.1. Pretrained variant.** Pretraining in greedy layer-wise manner was shown to be a possible way of improving performance [39]. The idea behind pretraining is to initialize the weights and biases of the model before training in order to avoid local minima due to random initialization. Therefore, pretraining is performed before the actual training of the model using the inputs.

The pretraining was done in the following manner:

1. Training a layer and saving the weights and biases for that layer–for example, the first layer contains 70 neurons with an input and output of 79, therefore it would result in 79x70 + 70x79 weights and 70 biases

2. Saving the codes resulted–for the previous example, codes were of size 70

3. For the pretraining of the next layer, the steps are repeated, but the input consists of the codes saved.

4. These steps are repeated until the desired code size is reached as the layer

5. The final step is to use all the weights and biases saved as the initial values for the training

**2.2.2. FT and WFT variants.** Using the coefficients of the Fourier integral, we can calculate the Fourier Spectrum, also known as the Fourier Transform (FT), where each function has a unique representation.

It is important to note that the FT of real functions, such as spikes, results in a frequency representationin the complex domain, with real and imaginary parts (or frequency and phase). To transform the inputs into the frequency domain, we have used the Fast Fourier Transform (FFT). Thus, many variants of FT AE can be created using any of the following frequency-domain descriptors: real part, imaginary part, magnitude, phase, power, various concatenations of the previous. In the evaluation section we only present the variant using the real part of the Fourier Transform as it had the best results.

**2.2.3. LSTM variant.** Long Short-Term Memory [40] (LSTM) are a type of Recurrent Neural Network (RNN). RNNs struggle with long-term dependencies, meaning that initial inputs will lose importance as newer inputs are shown. LSTMs, through their design, aim to better tackle this issue. LSTMs have a more complex chain structure; each LSTM module being composed of 4 simpler layers. The information in the LSTM state is regulated through gates in order to add or remove information. At least theoretically, for a large amount of spikes the LSTM architecture may be able to exceed simpler architectures in retaining the differences in waveforms between spikes from different neurons.

**2.2.4. Orthogonal variant.** An Orthogonal Autoencoder departs from the architecture of the autoencoder presented above. First, it constrains the weights of both encoder and decoder to be orthogonal by using a kernel regularizer. Second, it reduces correlation of the encoded features through the use of the covariance matrix by imposing a penalty on said features [42].

**2.2.5. Contractive variant.** The essence of the Contractive [43] variant is the loss function. It provides robustness to perturbations through regularization that favours the contracting of the training samples. This is achieved by keeping the weights small through regularization. It applies the squared Frobenius norm to the Jacobian of the function as the regularizer multiplied with a coefficient:

$$J_{CAE}(\theta) = \sum_{x \in D_n} (L(x, g(f(x)) + \lambda ||j_f(x)||_F^2)$$

**2.2.6. Preprocessing.** Alignment has to be applied as a first step of preprocessing before the execution of the feature extraction method. We have used the following formula, for multiple types of alignment at a chosen index:

$$new_{start_{spike}} = old_{start_{spike}} - (index_{align} - peak_{spike}) \tag{1}$$

Naturally, the point of start of a portion of the samples has to be changed; this is indicated in Formula (1) through the *new_start* and *old_start* terms. The *index* in Eq (1) represents the point to which all spikes will be shifted. Thus, we can choose to align the maximum peak of all spikes to the average index of the maximum peak across all samples, as shown in Fig 4C. Another, better option, is to align the amplitudes to the middle of the sample as it provides information about the spike from the perspective of both pre- and post-amplitude. The *peak* in Eq (1) represents the index at which the desired point of reference (typically, the peak) is found. For the alignment of the amplitude, it is the index of the maximum peak of each

sample. The formula permits the alignment of any point of reference, such as the minimum peak [44]. An analysis of the impact of alignment on the performance of PCA and the Autoencoders in shown in section 3.1.

In addition, we have applied two other preprocessing steps: scaling and shuffling. Due to the combination of the ReLU and 'tanh' activation function of the autoencoders, the reconstruction must be bounded within the [0,1] interval. As such, the data is also scaled to this interval in order to provide better reconstructions which allow for finer learning. The random shuffling of samples is required only for the synthetic datasets presented here, as the samples are ordered by their cluster affinity.

## 2.3. Data

**2.3.1. Synthetic datasets.** The validation of autoencoders was made by comparing the different variants of autoencoders with PCA, ICA and Isomap. The chosen datasets, 95 in number and denominated as simulations, originate from the Department of Engineering, University of Leicester UK and are publicly available. Each simulation is a dataset. The creation of these simulations was based on recordings from the neocortex of a monkey. They were generated using 594 different spike shapes [45]. The original study that introduces the simulations [45] also reviews different clustering algorithms and their results. Out of 20 different units, these algorithms were able to detect 10 in the best case.

The datasets were generated based on a real dataset recorded "in vivo". The waveform contains 316 points originally sampled at 96 KHz; afterwards this frequency was reduced to 24KHz, therefore 79 samples describe a spike. Being synthetic datasets, each of these spikes has a label, which allows for the use of external metrics to evaluate performance. Each simulation contains a multi-unit cluster, which is the noise, and a number of clusters that varies between 2 and 20. Each unique number of clusters has 5 simulations. Thus, there are 5 simulations with 2 clusters, 5 simulations with 3 clusters, and so on.

All but one of the clusters are single-units between 0 and 50μm away from the electrode. The firing rate follows a Poisson distribution with a mean between 0.1 and 2Hz. The amplitudes follow a normal distribution and have been scaled to values between 0.9 and 2 to simulate real data. No spikes with temporal overlapping are present in the data, such that spikes have at least 0.3ms between them.

The generated multi-unit cluster was added in order to increase the complexity of clustering for the tested algorithms. The simulated multi-unit contains 20 spike shapes, each of the 20 neurons firing being between 50–140μm away from the electrode. The amplitude of the spikes was fixed to 0.5, with an overall composite firing rate of 5Hz, with each of the 20 individual composing neurons having a firing rate mean of 0.25Hz following an independent Poisson distribution. Here, in order to increase clarity, the multi-unit cluster is always color-coded in white in all figures.

To evaluate the proposed approach in comparison with other state-of-the-art methods we have chosen the following 4 simulations out of the 95 available as they are representative of the issues that are present in feature extraction methods and allow for the evaluation of the methods on varying numbers of clusters covering a wide range and enabling a comprehensive evaluation of performance:

- Simulation 1 (Sim1—Fig 3A), containing 16 single-unit clusters and a multi-unit cluster (in total 17) with 12012 samples.

- Simulation 4 (Sim4—Fig 3B), containing 4 single-unit clusters and a multi-unit cluster (in total 5) with 5127 samples.

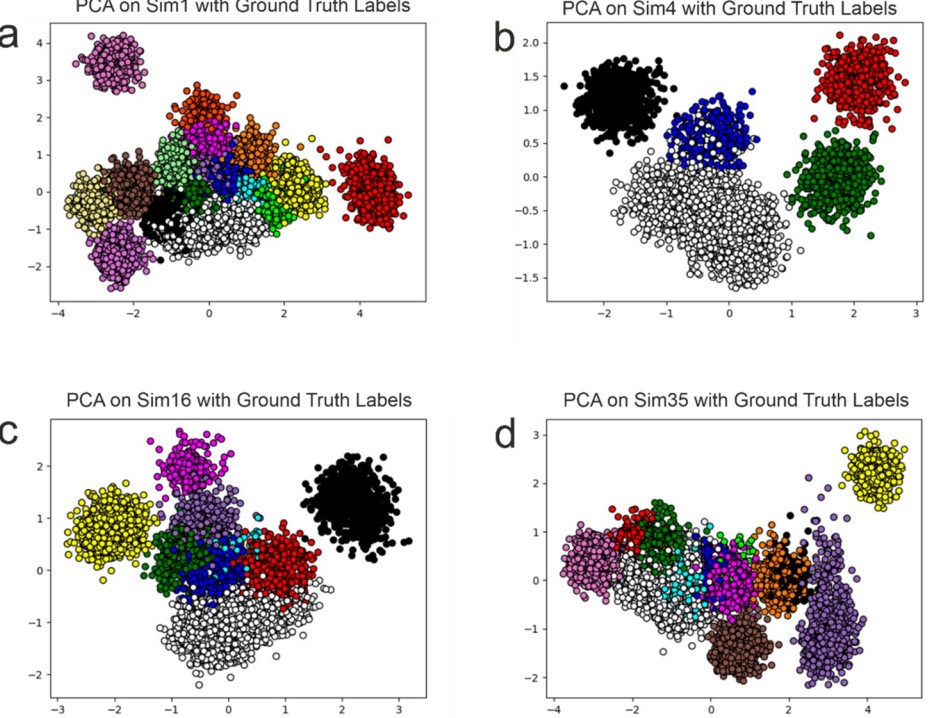

**Fig 3. PCA projection of the synthetic datasets.** PCA projection of 4 different simulations with distinct numbers of clusters; the colors represent the cluster assignment given in the ground truth.

- Simulation 16 (Sim16—Fig 3C), containing 8 single-unit clusters and a multi-unit cluster (in total 9) with 7556 samples.

- Simulation 35 (Sim35—Fig 3D), containing 12 single-unit clusters and a multi-unit cluster (in total 13) with 9481 samples.

- Simulation 14, containing 3 single-unit clusters and a multi-unit cluster (in total 4) with 4507 samples. This dataset was used for the visualization of the impact of alignment on feature extraction in section 3.1.

These simulations can also be viewed in Fig 3 through the use of PCA to reduce the dimensionality from 79 to 2. The overlapping clusters produced by PCA can be clearly seen in Fig 3, in none of the datasets is it able to perfectly separate all clusters.

**2.3.2. Real datasets.** The electrophysiological *"in vivo"* data was recorded from the brain of anaesthetized adult mice of the C57/B16 strain with A32-tet probes (NeuroNexus Technologies, Inc) at 32 kSamples /s (Multi Channel Systems MCS GmbH) during a visual stimulation. The stimuli were presented monocularly on a Beetronics 12VG3 12-inch monitor with a resolution of 1440x900, at 60fps and consisted of full-field drifting gratings (0.11 cycles/deg; 1.75 cycles/s; variable contrast 25–100%; 8 directions in steps of 45˚). The animals, on which the extracellular activity was recorded, were placed in the stereotaxic holder (Stoelting Co, Illinois, United States) and anaesthetized. Anesthesia was induced and maintained with isoflurane (ISO) in oxygen (5% for induction, 1–3% for maintenance). The heart rate, respiration rate, core body temperature, and pedal reflex were constantly monitored. A circular craniotomy (1x1 mm) was performed over the left visual cortex of the animal centred on 0–0.5 mm

anterior to lambda, 2–2.5 mm lateral to midline. To obtain multiunit activity (MUA) containing signals, the extracellular data was digitally filtered using a band-pass filter with a range of 300Hz-7000Hz using a bidirectional Butterworth IIR filter of order 3. An amplitude threshold, most commonly chosen between 3 and 5 [1] standard deviations of the recorded signal, was used to detect spike, which were then fed into the feature extraction algorithms. Spikes were identified as threshold crossings and subsequently used as input for the feature extraction algorithm.

Multiple datasets were accumulated from each animal over a period of 4 to 6h in order to minimise animal use. All experiments were performed in accordance with the European Communities Council Directive of 22 September 2010 (2010/63/EU) and approved by the Local Ethics Committee (3/CE/02.11.2018) and the National Veterinary Authority (147/04.12.2018).

## 2.4. Performance metrics

Six metrics were used for the validation of results: Adjusted Rand Index (ARI), Adjusted Mutual Information (AMI), V-Measure (VM), Calinski-Harabasz Score (CHS), Davies-Bouldin Score (DBS), and Silhouette Score (SS). The first three metrics are external metrics, while the last three are internal [46]. These are clustering performance metrics and are a suitable evaluation of the feature extraction due to the fact that spike sorting does not stop at this step but is followed by clustering. The external metrics provide information about the ability of the clustering algorithm to correctly identify the clusters based on a ground truth, which is heavily influenced by the separability offered through the feature extraction. This is due to the fact that with perfect separation, most clustering algorithms will be able to have a high performance. On the other hand, the internal metrics characterize the clustering based on the separability and shape of clusters, thus they are adequate for the evaluation of the feature extraction through the use of the ground truth labels for synthetic datasets. In fewer words, the internal metrics outline the properties of the clusters, while the external metrics evaluate the matching between the clustering and the ground truth. In Table 1, we present a short intuitive description and the range for each of the metrics.

We chose a multitude of evaluation metrics rather than an all-encompassing one, as they will appraise the performance from multiple considerations and perspectives. Thus, a method that provides greater performance across these numerous metrics is indicative of a balanced performance with an increased likelihood of an unbiased evaluation.

External metrics require the labels of the clustering algorithm, and the ground truth labels. Therefore, a clustering algorithm has to be applied after the feature extraction and we have

**Table 1. An intuitive description for each metric, its type and its range.**

| Name | Type | Description | Range [worst, best] |
|------|------|-------------|---------------------|
| ARI | External | Pair-by-pair comparison whether the points in the predicted cluster belong in the same true cluster | [−1, 1] |
| AMI | External | Mutual information based on entropy is used to calculate the agreement of true and predicted labels | [0, 1] |
| VM | External | Harmonic mean of conditional entropies between the true and predicted clusters | [0, 1] |
| DBS | Internal | Ratio of the inter-cluster and intra-cluster sum of squared distances | (Inf, 0] |
| CHS | Internal | The average of a function that evaluates inter-cluster distances and the size of the cluster | [0, Inf) |
| SS | Internal | Cluster quality is evaluated as the balance between a cluster's tightness and separation | [−1, 1] |

chosen K-Means [47]. K-means has a long history of use as a clustering algorithm and many variations have been developed. It was introduced in spike sorting in 1988 and remained the de facto clustering algorithm for a long time [48, 49]. Furthermore, newly developed spike sorting techniques and pipelines are based on it or use it [7, 50] and in recent evaluations K-Means has been shown to still be a highly performant option, as it placed third in the evaluation of 25 clustering algorithms [48].

K-Means is a partition-based clustering algorithm. It partitions the space into k partitions, where each sample is appointed to the closest centroid based on the Euclidean distance. K-Means has several disadvantages. First, it requires the number of clusters as an input which is hard to provide for real data. Second, in its most basic form it is not deterministic, such that each execution may result in a different clustering. Through recent optimizations it has been improved and has increased stability. Third, K-Means has difficulties in separating overlapping clusters. In our case, this is an advantage: If the performance of K-Means is higher for a certain feature extraction method it denotes that the method provides better separation.

ARI [51–53] (3) is an adjustment of the Rand Index (RI) metric in order to handle chances. ARI is an external clustering metric; therefore, it requires a ground truth for the dataset. RI [54] (2) makes comparisons between pairs of points to determine if it is an agreement, when the two points are in the same cluster for both the predicted and the true labels, or a disagreement, when they belong to different clusters. The formulas used to calculate the metric are the following:

$$RI = \frac{agreements}{agreements + disagreements} \tag{2}$$

$$ARI = \frac{RI - ExpectedRI}{MaxRI - ExpectedRI} \tag{3}$$

where *MaxRI* is the upper bound and *ExpectedRI* is the expected placement of pairs in the same class using the permutation model and calculated based on the contingency table [51].

AMI [52, 55] (5) is an adjustment of the Mutual Information (MI) metric through the use of entropy, denoted as H. Moreover, AMI also contains the normalization [52, 56, 57] of Normalized Mutual Information. MI (4) is calculated between two clusters U and V, where N is the size of the dataset and |X| is the number of points in subset X.

$$MI(U, V) = \sum_{i=0}^{|U|} \sum_{j=0}^{|V|} \frac{|Ui \cap Vj|}{N} \log \log \frac{N|Ui \cap Vj|}{|Ui||Vj|} \tag{4}$$

$$AMI = \frac{MI(U, V) - E(MI(U, V))}{average(H(U), H(V)) - E(MI(U, V))} \tag{5}$$

V-Measure [17] (6) is the harmonic mean of Homogeneity and Completeness. A cluster is considered to be homogeneous (7) when all the points of that cluster are part of the same class. By switching the predicted and true labels, completeness is obtained. Completeness (8) is achieved when all the points of a class are part of the same cluster. We have chosen beta equal

to 1 as given by the original formula [17].

$$V - Measure = \frac{(1 + beta) * Homogeneity * Completeness}{beta * Homogeneity + Completeness} \tag{6}$$

$$Homogeneity = 1 - \frac{H(C|K)}{H(C)} \tag{7}$$

$$Completeness = 1 - \frac{H(K|C)}{H(K)} \tag{8}$$

where H(C|K) is the conditional entropy of the true cluster given the predicted cluster, H(C) is the entropy of the true cluster, while H(K|C) is the conditional entropy of the predicted cluster given the true cluster and H(K) is the entropy of the predicted cluster.

All the metrics presented until this point are external metrics and require a ground truth to compare with the predicted labels. Furthermore, all these metrics have bounded scores in the [0, 1] interval with higher values being more desirable.

The following three metrics are internal and therefore do not require a ground truth to be used. The internal metrics were used with the ground truth labels for the evaluation of the synthetic datasets. These metrics evaluate the intra-cluster and inter-cluster distances and the morphology of the clusters producing an adequate evaluation of the feature extraction capabilities.

DBS [58–60] (10) finds the mean similarity between clusters, where similarity, denoted as $R$ (9), is defined by the distance between clusters and their sizes. The minimum value of this index is 0. The closer the result is to 0, the better separation exists between clusters. This may come as counterintuitive as it is the only metric where lower values represent a higher performance. The DBS metric is given by the following equations:

$$R_{i,j} = \frac{s_i - s_j}{d_{i,j}} \tag{9}$$

$$DBS = \frac{1}{k} \sum_{i=1}^{k} max\mathrm{R}_{i,j} \tag{10}$$

where $s_i$ is the mean of all distances between the points of cluster $i$ and its centroid, $d_{i,j}$ is the distance between clusters $i$ and $j$ given by their centroids, and $max(R_{i,j})$ is the maximum similarity of clusters $i$ and $j$.

CHS [17, 46] (11), also known as Variance Ratio Criterion, calculates the ratio between the intra-cluster and inter-cluster dispersion. Where $tr(X)$ denotes the trace of between cluster $Bk$ or within-cluster $Wk$ dispersion matrix, $n$ denotes the size of the dataset and $k$ the number of clusters. The dispersion is defined as the sum of squared distances. For this metric, a higher value indicates a better result.

$$CHS = \frac{tr(Bk)}{tr(Wk)} * \frac{n - k}{k - 1} \tag{11}$$

SS [17, 18] (12) is calculated by measuring the mean distance between a point and the rest of the points of that cluster and the mean distance between the point and all the points of the nearest cluster. The score is bound between [–1, 1] where -1 represents an incorrect clustering, 0 overlapping clusters, and 1 a dense clustering. SS aims for the standard concept of a cluster,

dense and well separated, therefore such cases will give a higher score. The equation of SS is the following:

$$SS = \frac{b - a}{\max(a, b)} \quad (12)$$

where $b$ denotes the average of all distances between a point in cluster $i$ and all points of the nearest cluster $j$, and $a$ the average of all distances between a point in cluster $i$ and all other points in the same cluster.

It is important to mention that although used in evaluation of spike sorting techniques [15, 16], accuracy is not a suitable performance metric. First, because spike sorting is unsupervised and accuracy requires labels. Second, neuronal data is imbalanced because of the various firing rates of individual neurons, and it is has been extensively shown that accuracy is not appropriate for evaluating tasks on imbalanced data [61–64]. Nevertheless, through the use of the chosen metrics, we are able to evaluate the separation and shapes of cluster using the internal metrics and the correctness of clustering using the created features using the external metrics.

## 3. Results and discussion

### 3.1. Impact of alignment

Alignment can heavily impact the performance of feature extraction methods by enabling standardization to the input and, thus, preventing the deformation of clusters. By aligning the spikes to a common point of reference, such as the amplitude (maximum peak), feature extraction methods are better able to learn because of intra-feature correlation. When a single feature describes different parts of the spikes across samples, it may not be able to learn adequately, deforming the clusters and increasing the risk of underclustering or overclustering. We define underclustering as the labelling of two or more true clusters as a single one by a clustering algorithm, while overclustering as a single true cluster being labelled as two or more clusters by the clustering algorithm. In Fig 4 we show the clusters of the ground truth, while overclustering and underclustering are defined as a comparison between a ground truth and the labelling of a clustering algorithm. Fig 4B shows an exceptional case, where the alignment of spikes completely averts theartitionning of the white cluster into two sub-clusters that will be identified as two separate clusters by most clustering algorithms. From a theoretical standpoint, this case will induce overclustering due to the visible separation between the samples of the same cluster, but more than that it shows that it is a poor extraction of features as these characteristics are not representative of the data demonstrating that alignment can have a high impact on the results. We have chosen to show this phenomenon with simulation 14, as the number of samples and clusters is low, and this permits a smooth visualization. The impact of alignment is clearly visible in Fig 4, while its effect on the performance of the feature extraction can be viewed in Table 2. We have highlighted the highest performances for each metric by bolding the values.

### 3.2. Analysis of parameters

We start the evaluation with the most conventional parameters of a neural network: the number of epochs and the learning rate. We assess the importance of each of these parameters in the learning process of an autoencoder and how different values may affect the performance.

We have chosen 3 values for the number of epochs: 50, 100 and 500. The impact of the number of epochs on the performance of the AE variant on Sim4 dataset can be viewed in Table 3. Increasing the number of epochs does not improve the performance from the

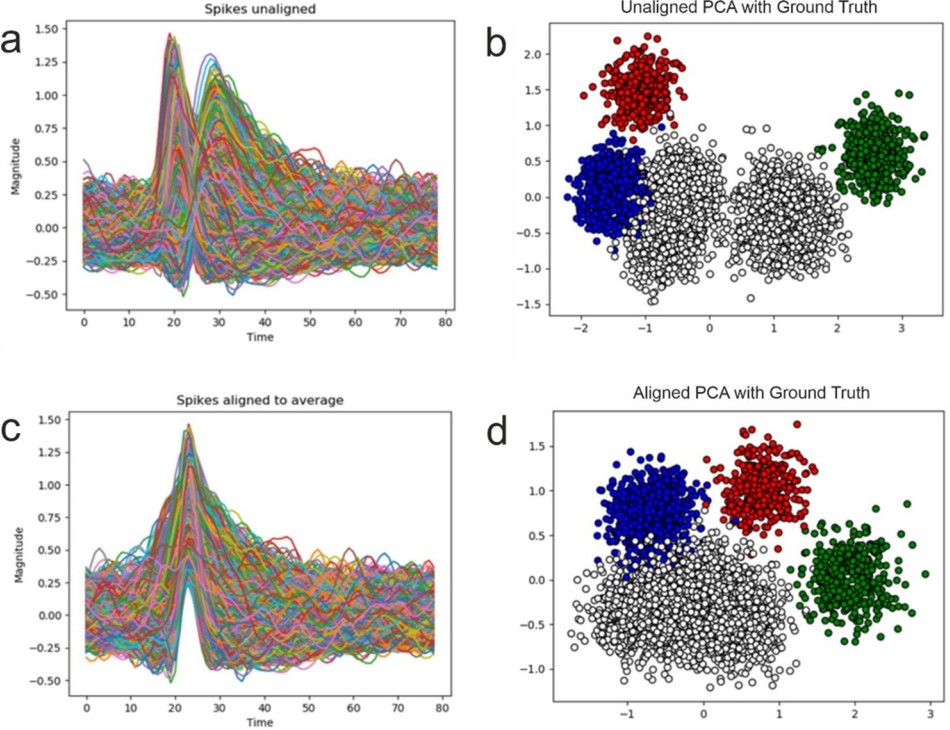

**Fig 4. Impact of alignment.** PCA applied on Sim14 with and without alignment. The white cluster is kept together but the overlap with the blue cluster remains.

**Table 2. Impact of alignment.** Comparison of different Alignments on the performance of PCA and AE.

|               | ARI  | AMI  | VM   | DBS  | CHS      | SS   |
|---------------|------|------|------|------|----------|------|
| PCA-Unaligned | 0.52 | 0.77 | 0.77 | **0.54** | 5,383.69 | 0.47 |
| PCA-Aligned   | 0.57 | 0.8  | 0.8  | 0.58 | 7,550.23 | 0.48 |
| AE-Unaligned  | 0.7  | 0.83 | 0.83 | 0.62 | 6,260.41 | 0.49 |
| AE-Aligned    | **0.98** | **0.96** | **0.96** | 0.55 | **7,807.64** | **0.61** |

**Table 3. Impact of the number of epochs through comparison of different number of epochs on the results of the AE variant on Sim4.** The metrics show that increasing the number of epochs does not necessarily increase the performance of the model as the best performance across all metrics is obtained at 50 epochs, the values indicating the highest performance for each metric have been bolded.

|                | ARI  | AMI  | VM   | DBS  | CHS       | SS   |
|----------------|------|------|------|------|-----------|------|
| Epochs = 50    | **0.98** | **0.97** | **0.97** | **0.27** | **52,012.95** | **0.79** |
| Epochs = 100   | 0.91 | 0.88 | 0.88 | 0.35 | 28,541.54 | 0.66 |
| Epochs = 500   | **0.98** | 0.96 | 0.96 | 0.55 | 7,807.64  | 0.61 |

perspective of the clustering (first three metrics), while it actually decreases the performance of the feature extraction (last three metrics). These results are in concordance with the loss values of the model during training that can be viewed in S1 Fig.

The impact of the learning rate on the performance of the AE variant on Sim4 dataset can be viewed in Table 4. At higher learning rates, the model is unable to learn or learns poorly, whilst lower values only slightly change the performance of the model.

**Table 4. Impact of learning rate comparison of different learning rates (LR) on the results of the AE variant on Sim4.** The metrics show that a lower learning rate offers higher performance across the evaluation metrics with one exception, the CHS that is close to the highest value obtained, the values indicating the highest performance for each metric have been bolded.

|  | ARI | AMI | VM | DBS | CHS | SS |
|---|---|---|---|---|---|---|
| LR = 0.1 | 0 | 0 | 0 | - | 0 | 0 |
| LR = 0.01 | 0.47 | 0.69 | 0.69 | 0.47 | **12,450.86** | 0.39 |
| LR = 0.001 | 0.98 | 0.96 | 0.96 | 0.55 | 7,807.64 | 0.61 |
| LR = 0.0001 | **0.98** | **0.97** | **0.97** | **0.43** | 12,343.35 | **0.62** |

## 3.3. Performance evaluation

The results of ICA and Isomap used in these evaluations have been chosen for each dataset by using a grid search. From our observations, we have found that the parameters of ICA do not heavily influence the results as the performance metrics change by a value smaller than 1e-3 with the exception of CHS that has no upper bound and we find differences of 1e-1. The highest performance is obtained when the number of iterations is chosen such that the algorithm reaches convergence. We have found that a lower tolerance offers slightly better results and saturates at about 1e-5 and the best performance is given by the 'logcosh' form.

Isomap has several parameters with many possible values. We have evaluated a range of values for each parameter. We have chosen to consider a number of neighbouring points instead of a radius for Isomap and have found that the best algorithm is KdTree, while the BallTree and brute approaches offer very close results. With regard to the distance function, we have found that the cosine distance offers the lowest performance, while the Minkowski (equivalent to the Euclidean distance for the 2D case) and Manhattan distances have very similar results with the Minkowski distance shining through by a degree of up to 1e-1. Similar to the ICA case, for the CHS metric it can have a higher impact, nevertheless smaller than 1 for most parameters except distance where differences of up to 1e4 can appear. We have found that the tolerance has a small impact on the results, smaller than 1e-3 and the best performance lies in a tolerance range of 1e-3 to 1e-5 depending on other parameters. The number of iterations chosen had no impact on the results as it was higher than the point of convergence. Regarding the shortest path, we have found that Floyd-Warshall and Dijkstra algorithms have almost identical results the difference appears on ARI and CHS of values smaller than 1e-4. For the eigenvalue decomposition, the direct and Arnoldi approaches have been tested and we have obtained similar performance depending on the other parameters.

**3.3.1. Performance evaluation of synthetic data.** The 95 synthetic datasets [45] contain varying numbers of clusters and spike shapes providing the complexity required for a comprehensive evaluation of the methods. In Fig 5, we present the results obtained for each metric across all 95 datasets for each method presented. A statistical analysis using t-tests with a Bonferroni correction can be examined in the S4 Fig and a ranking of the methods based on their performance for each metric using Borda rank aggregation [65] in S2 Table.

The 4 chosen synthetic datasets allow for the evaluation of the feature extraction methods' impact on the performance of the clustering algorithm and the ability of the feature extraction method to separate the clusters as the number of cluster increases. Furthermore, through this choice of datasets we are also able to evaluate the performance of the feature extraction methods on different numbers of clusters. The ranking of the methods for these datasets using the Borda rank [65] aggregation can be viewed in S2 Table. The evaluation offered by the performance metrics can be viewed for each dataset specifically in Tables 4–7 and they show consistency of performance for each autoencoder variant across datasets. A point to remember is that the DBS metric is inverted, in the sense that lower values indicate a better clustering. Each

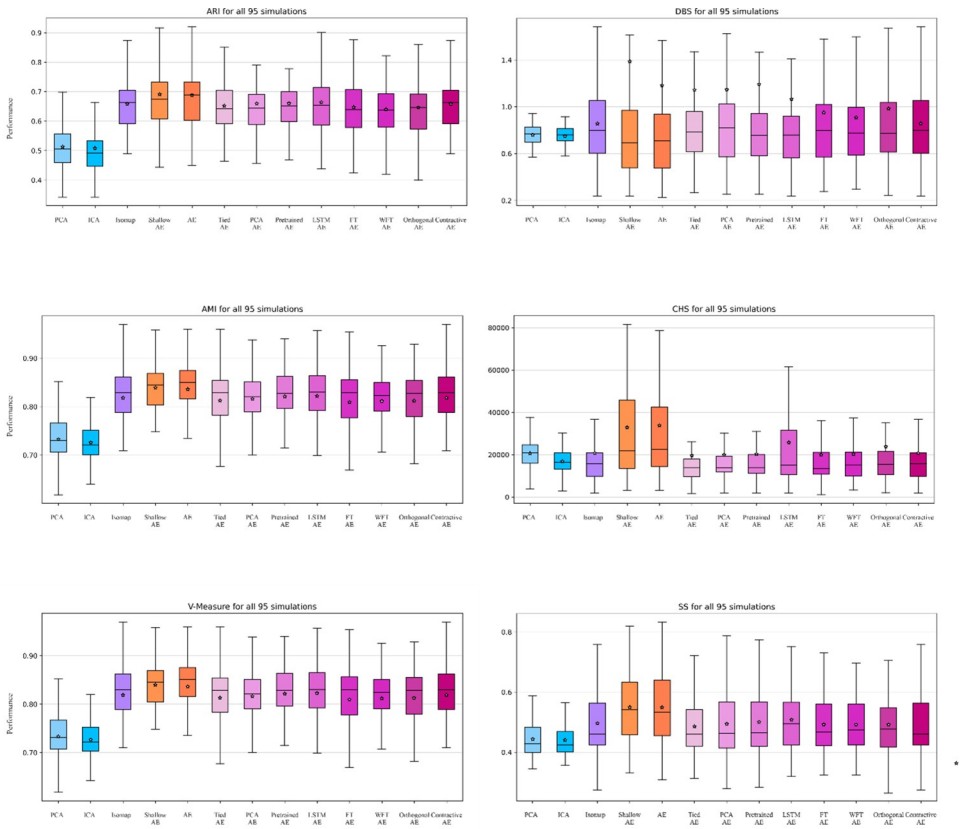

**Fig 5. Analysis of all methods on 95 synthetic datasets.** Performance evaluation of the classical feature extraction methods and the autoencoder variants for each metric on all 95 synthetic datasets, the star represents the average value.

of the synthetic datasets presented in this subsection contains a multi-unit cluster that has been color-coded as white across all figures. This cluster also contains the highest number of samples thus, offering imbalance to the clusters.

We start this evaluation with the easiest of the datasets chosen, containing 5 clusters, as it provides the best visualization of the performance of the feature extraction techniques. A common occurrence across the state-of-the-art methods is the superposition of the multi-unit cluster (white) with the same single-unit cluster (blue). This consistency highlights an inability of these methods to find the features that provide separation. The circumstance presented is clearly visible in Fig 6. The Isomap feature extraction method has the best performance out of the state-of-the-art methods, it is imperative to highlight that its high performance is also attributable to the ability of K-Means to correctly assign samples to clusters. As Fig 6C displays some overlap persisting.

The performance evaluation from the perspective of the 6 metrics is given in Table 5. We perceive the evaluation of the metrics to be correlated to the visual performance assessed from the plots presented in Fig 6. In the case of the simplest synthetic dataset, we find that four autoencoder variants have a greater performance: Shallow AE, AE, Pretrained AE, and Contractive AE. These are the precise variants that manage to separate the blue single-unit cluster from the white multi-unit cluster that state-of-the-art methods were unable to do. The best performance is offered by methods that manage to condense the samples of clusters into a dense group, united with samples of the same cluster and segregated from those of other clusters.

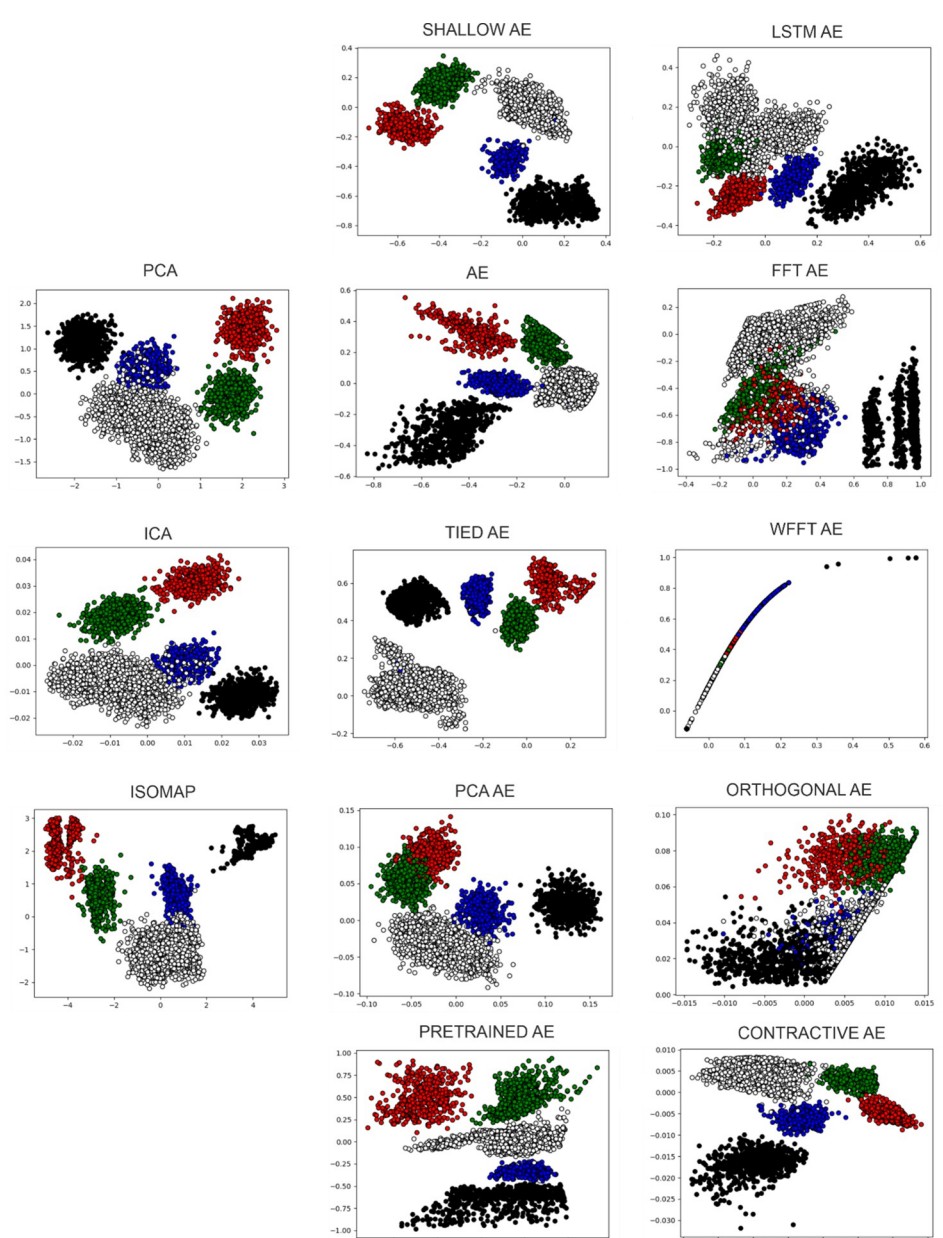

**Fig 6. Feature extraction of the Sim4 dataset.** All feature extraction methods applied on Sim4; the colors present the ground truth.

The Sim1 dataset is one of the more difficult ones. It presents a high number of clusters, 17 in number, that have some similarities in spike shapes resulting in a high amount of overlap between clusters. This is shown in both the plots displayed in Fig 7 as in the performance evaluations of Table 6. State-of-the-art methods are able to completely separate a low number of clusters. PCA and ICA only manage to separate 2 clusters, whilst the rest are coalesced into an amalgamation. Isomap provides the highest number of separated clusters by any of the state-of-the-art methods used, with 5 clusters separated, but the same tendency to join the rest of the clusters remains and it clearly visible in Fig 7. Whereas a part of the autoencoder variants, such as the Shallow AE and AE variations, are able to separate up to 10 clusters with minimal

**Table 5. Comparison of different autoencoder variations vs state-of-the-art methods on Sim4 (5 clusters), the optimal value of neighbours for Isomap was found as 90 through grid search.** The metrics indicate that several variations of autoencoders and the Isomap algorithm are able to outperform conventional algorithms such as PCA or ICA for the Sim4 dataset that contains a low number of clusters, the values indicating the highest performance for each metric have been bolded.

| | ARI | AMI | VM | DBS | CHS | SS |
|---|---|---|---|---|---|---|
| PCA | 0.57 | 0.8 | 0.8 | 0.58 | 7,550.23 | 0.48 |
| ICA | 0.61 | 0.8 | 0.8 | 0.53 | 7,000.8 | 0.51 |
| Isomap | 0.96 | 0.95 | 0.95 | 0.47 | 18,588.430 | 0.61 |
| Shallow AE | **0.99** | **0.98** | **0.98** | 0.43 | 13,835.92 | 0.65 |
| AE | 0.98 | 0.96 | 0.96 | 0.55 | 7,807.64 | 0.61 |
| Tied AE | 0.59 | 0.8 | 0.8 | 0.44 | 13,698.71 | 0.68 |
| PCA AE | 0.59 | 0.79 | 0.79 | 0.65 | 8,959.50 | 0.51 |
| Pretrained AE | 0.91 | 0.92 | 0.92 | **0.27** | **33,910.79** | **0.73** |
| LSTM AE | 0.31 | 0.59 | 0.59 | 0.82 | 3,592.13 | 0.25 |
| FT AE | 0.36 | 0.51 | 0.51 | 1.63 | 2,672.27 | 0.18 |
| WFT AE | 0.43 | 0.59 | 0.59 | 9.13 | 2,293.02 | 0.28 |
| Orthogonal AE | 0.32 | 0.44 | 0.44 | 8.71 | 4,866.81 | 0.05 |
| Contractive AE | 0.97 | 0.95 | 0.95 | 0.46 | 14,852.24 | 0.62 |

overlap between the rest. This is correlated to the performance estimation of the evaluation metrics, presented in Table 6. Notably, the same variants, as in the previous case, show the best performance.

We presented the evaluation of the most extreme cases, 5 and 17 clusters. To investigate how the number of clusters affect these methods, in the next cases we assess the performance of these methods with regard to the intermediate cases of 9 and 13 clusters, presented respectively in Tables 6 and 7. It is noticeable that the exact same autoencoder variants surface above the state-of-the-art methods regardless of the number of clusters. In these latter cases, the LSTM and Tied AE variants show an increased performance as well. Nonetheless, their performance remains closer to that of the state-of-the-art methods although consistent across these synthetic datasets.

**Table 6. Comparison of different autoencoder variations vs state-of-the-art methods on Sim1 (17 clusters), the optimal value of neighbours for Isomap was found as 110 through grid search.** The metrics indicate that several variations of autoencoders and the Isomap algorithm are able to outperform conventional algorithms such as PCA or ICA for the Sim1 dataset that contains a high number of clusters. Moreover, two autoencoder variants outperform Isomap on 5 out of the 6 metrics, the values indicating the highest performance for each metric have been bolded.

| | ARI | AMI | VM | DBS | CHS | SS |
|---|---|---|---|---|---|---|
| PCA | 0.5 | 0.74 | 0.74 | 1 | 12,020.59 | 0.26 |
| ICA | 0.48 | 0.72 | 0.73 | 1.05 | 9,929.23 | 0.24 |
| Isomap | 0.62 | 0.83 | 0.83 | 0.61 | **59,532.32** | 0.51 |
| Shallow AE | 0.75 | **0.92** | **0.92** | **0.48** | 28,238.61 | 0.57 |
| AE | **0.81** | 0.88 | 0.88 | 0.70 | 17,491.77 | 0.48 |
| Tied AE | 0.61 | 0.81 | 0.81 | 2.73 | 5,913.32 | 0.35 |
| PCA AE | 0.4 | 0.67 | 0.67 | 1.99 | 8,416.4 | 0.14 |
| Pretrained AE | 0.68 | 0.85 | 0.85 | 0.66 | 17,011.47 | 0.45 |
| LSTM AE | 0.60 | 0.83 | 0.83 | 1.13 | 17,221.50 | 0.34 |
| FT AE | 0.31 | 0.58 | 0.58 | 2.56 | 6,807.19 | 0.01 |
| WFT AE | 0.38 | 0.65 | 0.65 | 3.22 | 8,507.47 | 0.13 |
| Orthogonal AE | 0.27 | 0.53 | 0.53 | 3.36 | 5,788.95 | 0.00 |
| Contractive AE | 0.71 | 0.87 | 0.87 | 0.70 | 57,764.18 | **0.59** |

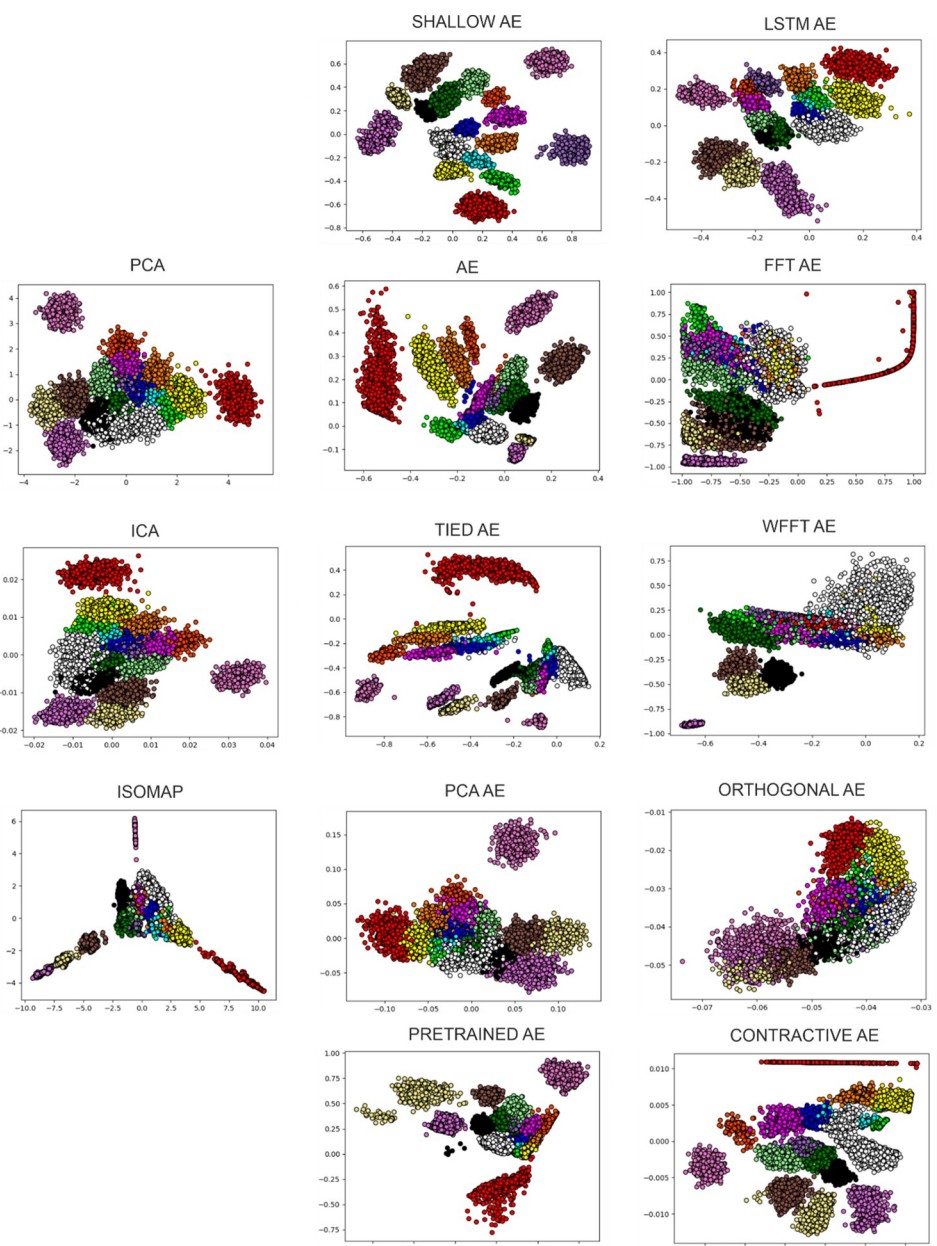

**Fig 7. Feature extraction of the Sim1 dataset.** All feature extraction applied on Sim1; the colors present the ground truth.

**3.3.2. Performance evaluation of real data.** The real dataset was recorded extracellularly using a silicone probe that provided 32 channels. Out of the 32 channels, we have chosen four that had an adequate number of spikes. The results of the autoencoder variants and the state-of-the-art feature extraction algorithms on real data, specifically on channel 17, is shown in Fig 8. As previously stated, real data contains no ground truth. From this irrefutable reality, the following implication emerges: We are not able to evaluate the performance of feature extraction separately from that of the clustering. Moreover, out of the metrics presented only the internal metrics remain available. As such, the colours will now represent the labelling of the K-Means clustering algorithm. Thus, the colours will not be corresponding across different

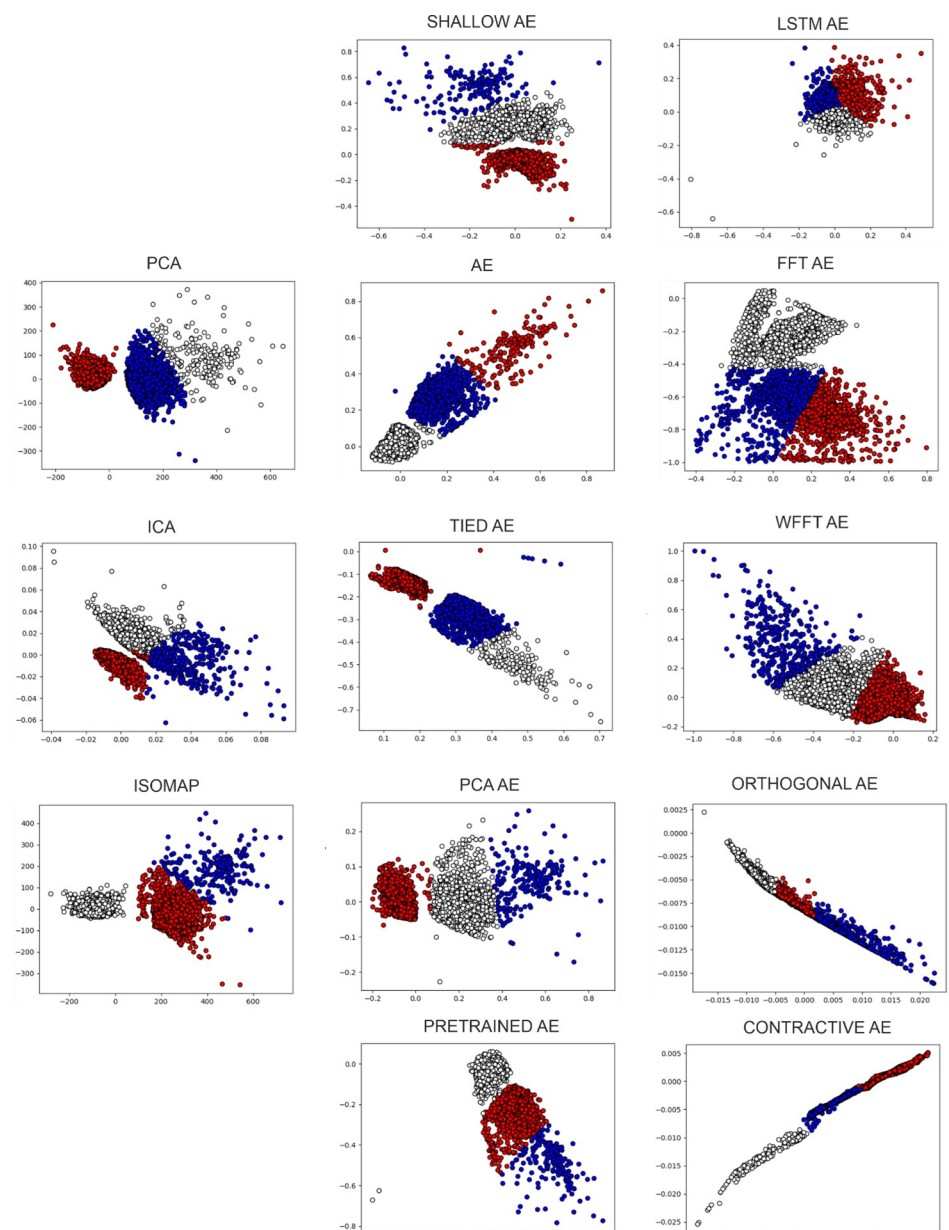

**Fig 8. Feature extraction of real data.** All feature extraction applied on real data channel 17; the colors present the labelling of K-Means.

plots. In this section, we present the performance of the feature extraction methods on all 4 channels in Tables 8–11. A point to remember is that the DBS metric is inverted, in the sense that lower values represent a better clustering.

The real data of channel 17 visually presents 3 separable clusters in most feature extraction methods. This can be viewed in Fig 9. Through empirical analysis of the spike shapes, we concur that this is most likely. Here, we analyze the ability of the feature extraction method to offer easily separable clusters for the clustering algorithm and the results can be viewed in Table 9. The state-of-the-art methods have a similar performance with one of the dense clusters separated from the overlap between the other dense cluster and the sparser one as shown

**Table 7. Comparison of different autoencoder variations vs state-of-the-art methods on Sim16 (9 clusters), the optimal value of neighbours for Isomap was found as 20 through grid search.** The metrics indicate that several variations of autoencoders and the Isomap algorithm are able to outperform conventional algorithms such as PCA or ICA for the Sim16 dataset that contains a moderate number of clusters. Moreover, two of autoencoder variants outperform Isomap with regard to the external metrics implying that these variants offer more separable features for the clustering algorithm, the values indicating the highest performance for each metric have been bolded.

| | ARI | AMI | VM | DBS | CHS | SS |
|---|---|---|---|---|---|---|
| PCA | 0.52 | 0.73 | 0.73 | 0.86 | 7,641.33 | 0.34 |
| ICA | 0.52 | 0.73 | 0.73 | 0.85 | 7,519.25 | 0.36 |
| Isomap | 0.65 | 0.80 | 0.80 | **0.58** | **35,485.69** | 0.51 |
| Shallow AE | **0.91** | **0.91** | **0.91** | 0.62 | 9,302.68 | 0.51 |
| AE | 0.63 | 0.81 | 0.81 | 0.72 | 1,0556.89 | 0.47 |
| Tied AE | 0.61 | 0.83 | 0.83 | 0.68 | 6,938.12 | 0.45 |
| PCA AE | 0.53 | 0.69 | 0.69 | 0.93 | 5,446.97 | 0.27 |
| Pretrained AE | 0.91 | 0.90 | 0.90 | 0.79 | 6,900.35 | 0.42 |
| LSTM AE | 0.60 | 0.82 | 0.82 | 1.05 | 9,392.85 | 0.43 |
| FT AE | 0.28 | 0.46 | 0.47 | 2.51 | 1,304.79 | -0.01 |
| WFT AE | 0.43 | 0.63 | 0.63 | 1.34 | 5,956.58 | 0.22 |
| Orthogonal AE | 0.46 | 0.58 | 0.58 | 1.65 | 6,244.37 | 0.27 |
| Contractive AE | 0.65 | 0.81 | 0.81 | 2.06 | 21,639.71 | **0.63** |

by Fig 9. The autoencoder variants have a similar performance visually, while in reality, the clusters provided by the more performing variants are more compacted, and the mean of the sparse cluster is further away. These are the reasons that a difference in performance appears. The separation of waveforms by clusters is presented in Fig 8 for the classical approaches and the AE variant. Notably, in contrast to the evaluation of the synthetic datasets, the performance of the LSTM AE that has been similar to state-of-the-art has dropped. Whilst, in addition to the four variants that performed well in the synthetic cases, here we also include the Tied and PCA variants within the highlighted ones. The Borda ranking [65] of the performance across the real data is presented in S3 Table.

Channel 4 has a similar distribution to that of channel 17, it contains 3 clusters identified from the spike shapes that are correlated with the results provided by the feature extraction

**Table 8. Comparison of different Autoencoder variations vs state-of-the-art methods on Sim35 (13 clusters), the optimal value of neighbours for Isomap was found as 20 through grid search.** The metrics indicate that several variations of autoencoders and the Isomap algorithm are able to outperform conventional algorithms such as PCA or ICA for the Sim35 dataset that contains a moderate number of clusters. Moreover, the Shallow AE variant outperforms Isomap with regard to the all metrics but CHS, the values indicating the highest performance for each metric have been bolded.

| | ARI | AMI | VM | DBS | CHS | SS |
|---|---|---|---|---|---|---|
| PCA | 0.45 | 0.69 | 0.69 | 1.4 | 11,156.65 | 0.18 |
| ICA | 0.4 | 0.66 | 0.66 | 1.92 | 6,464.01 | 0.15 |
| Isomap | 0.70 | 0.86 | 0.86 | 0.65 | **56,296.18** | 0.52 |
| Shallow AE | 0.79 | **0.94** | **0.94** | **0.51** | 13,879.97 | 0.56 |
| AE | 0.75 | 0.88 | 0.88 | 0.77 | 12,244.32 | 0.49 |
| Tied AE | 0.69 | 0.85 | 0.85 | 0.79 | 25,253.25 | 0.49 |
| PCA AE | 0.48 | 0.70 | 0.70 | 1.23 | 6,726.07 | 0.23 |
| Pretrained AE | **0.89** | 0.88 | 0.88 | 0.93 | 13,948.75 | 0.51 |
| LSTM AE | 0.69 | 0.86 | 0.86 | 0.69 | 17,022.40 | 0.45 |
| FT AE | 0.36 | 0.57 | 0.57 | 5.82 | 4,056.02 | 0.05 |
| WFT AE | 0.48 | 0.65 | 0.65 | 4.32 | 6,951.03 | 0.13 |
| Orthogonal AE | 0.32 | 0.60 | 0.60 | 1.80 | 19,288.69 | 0.03 |
| Contractive AE | 0.72 | 0.88 | 0.88 | 0.55 | 55,588.48 | **0.58** |

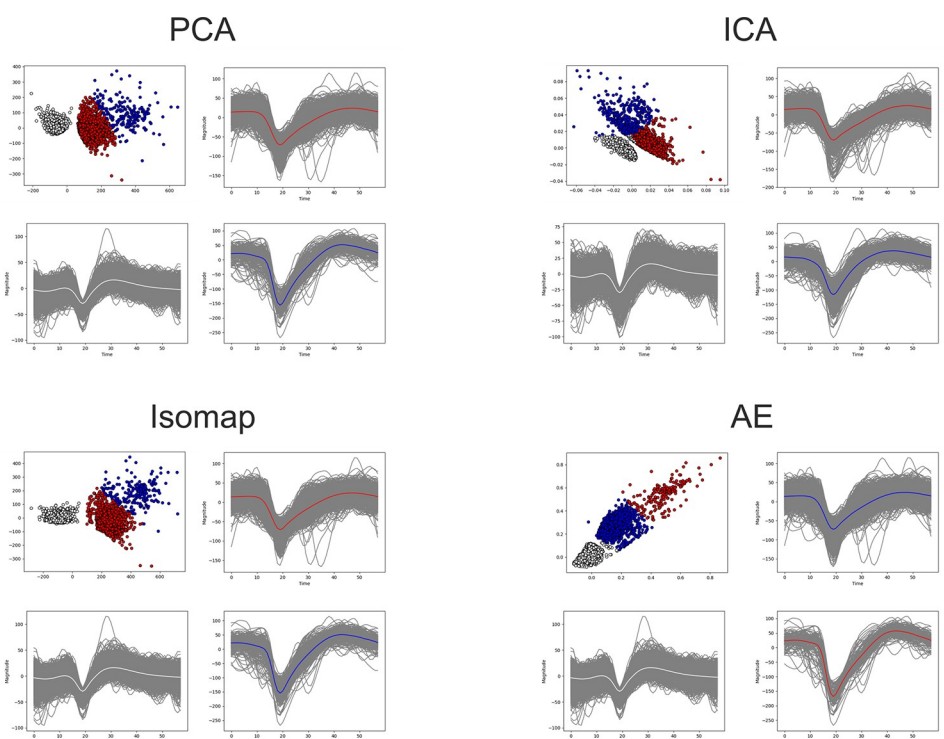

**Fig 9. The features and the waveform separation by colors for the classical methods (PCA, ICA, Isomap) and the AE variant.**

methods. It contains two dense clusters and one sparse cluster. The performance evaluation is presented in Table 10. The performance of autoencoder variants for this channel remains similar to that of state-of-the-art methods with the exception of the AE and Orthogonal AE variants.

**Table 9. Comparison of different autoencoder variations vs state-of-the-art methods on real data channel 17, the optimal value of neighbours for Isomap was found as 20 through grid search.** The metrics indicate that the AE variant is able to outperform the conventional feature extraction algorithm with the exception of the CHS metric, while the performance of the other variants varies across metrics due to the fact that the internal metrics evaluate the quality of the clustering from different perspectives as no ground truth is available for real data. The values indicating the highest performance for each metric have been bolded.

|  | DBS | CHS | SS |
|---|---|---|---|
| PCA | 0.63 | 20,637.96 | 0.76 |
| ICA | 0.72 | 8,200 | 0.64 |
| Isomap | 0.45 | **68,339.47** | 0.86 |
| Shallow AE | 0.65 | 14,000 | 0.72 |
| AE | 0.24 | 14,100 | **0.95** |
| Tied AE | 0.45 | 34,700 | 0.8 |
| PCA AE | **0.19** | 52,767.79 | 0.87 |
| Pretrained AE | 0.32 | 45,700 | 0.92 |
| LSTM AE | 1 | 3,844.78 | 0.38 |
| FT AE | 0.84 | 12,555.42 | 0.61 |
| WFT AE | 0.79 | 8,700 | 0.45 |
| Orthogonal AE | 0.58 | 20,500 | 0.58 |
| Contractive AE | 0.36 | 31,881.83 | 0.8 |

**Table 10. Comparison of different autoencoder variations vs state-of-the-art methods on real data channel 4, the optimal value of neighbours for Isomap was found as 30 through grid search.**

|  | DBS | CHS | SS |
|---|---|---|---|
| PCA | 0.61 | 10,278.06 | 0.75 |
| ICA | 0.61 | 5,601.31 | 0.66 |
| Isomap | 0.46 | 19,764.71 | 0.81 |
| Shallow AE | 0.56 | 11,659.44 | 0.71 |
| AE | 0.47 | **21,788.67** | **1** |
| Tied AE | 0.53 | 9,510 | 0.72 |
| PCA AE | 0.68 | 7,084.1 | 0.7 |
| Pretrained AE | 0.57 | 7,424.04 | 0.7 |
| LSTM AE | 0.92 | 2,680 | 0.4 |
| FT AE | 0.79 | 3,340 | 0.47 |
| WFT AE | 0.74 | 4,470 | 0.49 |
| Orthogonal AE | **0.38** | 3,530 | 0.93 |
| Contractive AE | 0.51 | 16,100 | 0.71 |

Channel 6 contains 4 clusters identified from the spike shapes that are correlated with the results provided by the feature extraction methods. Each identified spike shape includes a similar number of samples; thus it is expected to have clusters with similar distributions. The performance evaluation is presented in Table 11. For this channel, only the AE variant has been deemed to have an undeniably better performance than state-of-the-art methods. Other variants present a similar performance to that of state-of-the-art techniques with varyingly slightly better or worse results across different metrics.

Channel 26 contains 4 clusters identified from the spike shapes that are correlated with the results provided by the feature extraction methods. It contains two dense clusters, one sparse cluster and an intermediate one. The performance evaluation is presented in Table 12. Within the case of channel 26, multiple variants show an increased performance to that of state-of-the-art techniques. Most notable is the AE variant that has the best results even in comparison

**Table 11. Comparison of different autoencoder variations vs state-of-the-art methods on real data channel 6, the optimal value of neighbours for Isomap was found as 30 through grid search.** The metrics indicate that for this dataset the AE variant is able to outperform the conventional feature extraction algorithm with regard to all metrics while the other variants have similar performances to that of PCA and ICA. The values indicating the highest performance for each metric have been bolded.

|  | DBS | CHS | SS |
|---|---|---|---|
| PCA | 0.63 | 15,800 | 0.62 |
| ICA | 0.74 | 11,500 | 0.56 |
| Isomap | 0.56 | 24,768.1 | 0.66 |
| Shallow AE | 0.59 | 14,300 | 0.61 |
| AE | **0.4** | **68,500** | **0.73** |
| Tied AE | 0.61 | 17,100 | 0.63 |
| PCA AE | 0.62 | 16,500 | 0.61 |
| Pretrained AE | 0.75 | 12,000 | 0.6 |
| LSTM AE | 0.64 | 5,970 | 0.52 |
| FT AE | 0.84 | 6,414.04 | 0.46 |
| WFT AE | 0.9 | 7,330 | 0.35 |
| Orthogonal AE | 0.6 | 16,000 | 0.58 |
| Contractive AE | 0.49 | 58,600 | 0.66 |

**Table 12. Comparison of different autoencoder variations vs state-of-the-art methods on real data channel 26, the optimal value of neighbours for Isomap was found as 30 through grid search.** The metrics indicate that for this dataset the AE variant is able to outperform the conventional feature extraction algorithm with regard to all metrics while the other variants have similar performances to that of PCA, ICA and Isomap. The values indicating the highest performance for each metric have been bolded.

|  | DBS | CHS | SS |
|---|---|---|---|
| **PCA** | 0.81 | 2,110 | 0.55 |
| **ICA** | 0.77 | 1,550 | 0.42 |
| **Isomap** | 0.78 | 3,118 | 0.44 |
| **Shallow AE** | 0.81 | 3,320 | 0.6 |
| **AE** | **0.42** | **12,500** | **0.74** |
| **Tied AE** | 0.68 | 3,812 | 0.56 |
| **PCA AE** | 0.8 | 2,090 | 0.51 |
| **Pretrained AE** | 0.73 | 2,860 | 0.54 |
| **LSTM AE** | 0.59 | 6,800 | 0.54 |
| **FT AE** | 0.59 | 4,180 | 0.52 |
| **WFT AE** | 0.88 | 1,120 | 0.38 |
| **Orthogonal AE** | 0.83 | 1,770 | 0.38 |
| **Contractive AE** | 0.68 | 4,230 | 0.54 |

with other variants. Nonetheless, several variants manage to outperform the state-of-the-art methods with regard to all evaluation metrics.

The performance of independent variants varies across the real channels. Nevertheless, the Shallow AE and AE variants provide consistently higher performance results across both synthetic and real data.

## 4. Conclusions

Here, we introduced several variants of Autoencoders as a feature extraction algorithm for spike sorting and investigated their performance by comparing them to conventional feature extraction algorithms used in spike sorting. Our focus was that of neural data, which has its particular characteristics that bestow complexity upon the problem of spike sorting [1].

We evaluated the autoencoder variants against state-of-the-art algorithms and assessed their performance using various performance evaluation metrics on multiple datasets. Our evaluation of the 95 synthetic datasets shows consistently that the autoencoder values have a higher performance according to the external metrics when compared to PCA or ICA while having similar values to Isomap. The internal metrics indicate that the autoencoder variants have a higher variability across the 95 synthetic datasets than the classical methods, nevertheless they show a higher performance overall.

The Autoencoder based variants can provide more separability for the clustering of neuronal spikes as shown visually and assessed through multiple performance evaluation metrics. Similarly, to the state-of-the-art methos, they do not require prior knowledge of the most informative features. One out of the ten variants presented in this study have shown a consistently higher performance as a feature extraction method within spike sorting for both synthetic and real datasets when used with K-Means, namely the AE variant. This is indicated in our evaluations of the metrics and the ranking of the methods presented in the Supporting Information section. Other autoencoder variants have proven to be situational, the Shallow variant appears to have a high performance on synthetic datasets, we hypothesize that this happens because synthetic spikes are simpler than the ones of real data. Thus, for real data more layers are required to separate the spikes as they are more complex. In fact, the AE variant is a

more complex architecture with more layers and it has a better performance than the Shallow AE variant on real data.

Naturally, the variability of a random initialization in K-Means can result in a higher or lower performance. We have chosen to use the k-means++ initialization and we present its variability across 1000 executions in the S2 Fig. Nevertheless, in certain cases even separated clusters can result in a poor clustering by K-Means, this can be seen in certain examples where the internal metrics are high, while the external metrics remain low. This is indicative of a faulty labelling by K-Means for some clusters. Furthermore, the application of these models does not require parameter tuning for a new dataset and as such, allows for a wide use within spike sorting with a feasibly higher performance. The Tied and Contractive variants also fair better on real datasets than the Shallow variant which is arguably the best on synthetic datasets. The Fourier variants (FT AE and WFT AE) learn the characteristics of the data in the frequency domain and have proven to be unable to find informative features. Nonetheless, as any neural network, autoencoders present some variability in their results, we provide an evaluation of the variability of the AE variant for 100 executions in the S3 Fig. PCA and ICA are ranked most commonly in the lower half, while Isomap, a more complex approach, places below the AE variant while its placement regarding other variants depends upon the dataset. Nevertheless, Isomap obtains high values for the CHS metric indicating that the extracted features create dense and separated clusters.

Autoencoders being an inherently unsupervised model are an adequate solution for spike sorting, while at the same time entailing no additional cost for data labelling. By their inner machinations, no segregation between training and testing datasets is needed. The training of the model on a dataset optimises it and the latent feature space can be extracted. Thus, these models are not affected by performance drop that can appear by an improper training as the whole dataset is used, consolidating its robustness. Notwithstanding, these autoencoders models require training for each new datasets, thus incurring the supplementary costs of training the model. The training of the model is dependent upon the number of samples in the dataset and the number of epochs. We deem these additional costs to be equitable with the increase in performance. The training of the deepest model with a total of 17 layers containing 12012 samples has been measured at an average of 0.351s per epoch, thus a training of 50 epochs would require only 17.57s. These evaluations were run on a laptop with AMD Ryzen 9 5900HX at 3.30GHz with 8 cores hyperthreaded, 32GB of RAM at 3200MHz, NVIDIA RTX3080 with 12GB of VRAM, 2TB SSD.

The decrease in performance with the increase of the number of clusters is a familiar difficulty within spike sorting. Even though, the autoencoder variants presented here provide a higher performance than state-of-the-art methods, the same phenomenon can be viewed throughout the evaluation of the external metrics of the synthetic datasets. However, it is noticeable that this does not occur within the evaluation of internal metrics for autoencoders, while the decrease remains for the state-of-the-art methods. We postulate that this phenomenon indicates that autoencoders can provide separation even with a higher number of clusters. However, further study is needed to assess the prevalence of this occurrence.

We conclude that autoencoders have a definite place in spike sorting, as they can outperform other feature extraction methods on all the criteria used in this evaluation. They allow for a high number of variants that have specific propensities that may be useful for datasets with certain characteristics. Furthermore, with the development of hardware [5] and software, data complexity and quantity will increase but so will the computational power, enabling autoencoders to tackle such data.

## Supporting information

**S1 Fig. Model loss during training.** Loss values during the training of the model on Sim4 for an AE model.
(TIF)

**S2 Fig. K-Means variability.** Variability of the used K-Means implementation for 1000 executions on the features provided by PCA using k-means++ initialization.
(TIF)

**S3 Fig. Autoencoder variability.** Variability of the shallow autoencoder on the Sim1 dataset (17 clusters) over 100 executions, where the white star represents the average score over 100. An important note is that DBS has lower scores for a higher performance and as DBS and CHS have only a lower bound they have been scaled between 0 and 1 by dividing with the maximum value across all executions of the autoencoder and the other methods.
(TIF)

**S4 Fig. Performance t-tests.** P value of t-tests (with a Bonferroni correction) for each of the metric on all 95 simulations.
(TIF)

**S1 Table. Methods ranking on a subset of synthetic data.** Borda rank aggregation of results for each metric on the 4 synthetic datasets (simulations 1, 4, 16, 35).
(DOCX)

**S2 Table. Methods ranking on synthetic data.** Borda rank aggregation of the results for each metric on all 95 synthetic datasets.
(DOCX)

**S3 Table. Methods ranking on real data.** Borda rank aggregation of the results for each metric on real data.
(DOCX)

## Acknowledgments

We want to thank our colleague Vlad Vasile Moca for insightful comments and discussions.

## Author Contributions

**Conceptualization:** Eugen-Richard Ardelean, Andreea Coporîie.

**Data curation:** Eugen-Richard Ardelean, Ana-Maria Ichim.

**Formal analysis:** Eugen-Richard Ardelean.

**Funding acquisition:** Raul Cristian Mureşan.

**Investigation:** Eugen-Richard Ardelean.

**Methodology:** Eugen-Richard Ardelean, Andreea Coporîie.

**Project administration:** Mihaela Dînşoreanu, Raul Cristian Mureşan.

**Resources:** Ana-Maria Ichim.

**Software:** Eugen-Richard Ardelean, Andreea Coporîie.

**Supervision:** Mihaela Dînşoreanu, Raul Cristian Mureşan.

**Validation:** Eugen-Richard Ardelean.

**Visualization:** Eugen-Richard Ardelean, Andreea Coporîie.

**Writing – original draft:** Eugen-Richard Ardelean.

**Writing – review & editing:** Eugen-Richard Ardelean, Andreea Coporîie, Ana-Maria Ichim, Mihaela Dînşoreanu, Raul Cristian Mureşan.

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
