## [Decision Letter · Decision Letter 0]

23 Nov 2022

PONE-D-22-25738A Study of Autoencoders as a Feature Extraction Technique for Spike SortingPLOS ONE

Dear Dr. Ardelean,

Thank you for submitting your manuscript to PLOS ONE. After careful consideration, we feel that it has merit but does not fully meet PLOS ONE’s publication criteria as it currently stands. Therefore, we invite you to submit a revised version of the manuscript that addresses the points raised during the review process.

ACADEMIC EDITOR: The opinions given by both reviewers are sharp and the author needs to take them seriously. The author needs to revise carefully.

We look forward to receiving your revised manuscript.

Kind regards,

Yiming Tang, Ph.D.

Academic Editor

PLOS ONE

Journal Requirements:

Reviewers' comments:

Reviewer's Responses to Questions

**Comments to the Author**

1. Is the manuscript technically sound, and do the data support the conclusions?

Reviewer #1: No

Reviewer #2: Partly

2. Has the statistical analysis been performed appropriately and rigorously? 

Reviewer #1: No

Reviewer #2: No

3. Have the authors made all data underlying the findings in their manuscript fully available?

Reviewer #1: Yes

Reviewer #2: No

4. Is the manuscript presented in an intelligible fashion and written in standard English?

Reviewer #1: No

Reviewer #2: Yes

5. Review Comments to the Author

Reviewer #1: This research failed to identify the problem and purpose of the study; hence, its novelty was minimal. In addition, the manuscript was poorly organized, thus I would recommend that the authors revise and resubmit it. In the event of resubmission, the following are some of my suggestions: Please adhere to the correct template, conduct additional literature reviews, compare your results to earlier work, and use standard citation format. Hope for the best.

Reviewer #2: Overall the work is interesting and useful. The proposed methods show promise. The evaluation methods are fairly rigorous, but can be improved by looking at dependence on parameter tuning and computing statistics. The presentation needs to provide some missing details. It was fun to read a paper from Transylvania on Halloween.

57: If the authors want to be more specific here, the problem isn't so much an imbalance, (a known imbalanced distribution can be handled) but unknown distribution, since changing firing rates mean different relative frequencies at different times.

59: Perhaps the authors can discuss the concept of single units and multiunits here and state what they mean when they write neurons in the manuscript.

61: Depending on the implementation, the 4th step is sometimes an offline (batch mode) clustering step, or alternatively an online classification step (using parameters previously found in offline batch mode clustering or by hand). This is a minor detail the authors can choose to include to be thorough.

100: Around here would be a nice place for a brief discussion of why autoencoders are well suited for the spike sorting enterprise.

156: Consider mentioning wavelet transform (Yuan etal 10.1016/j.jneumeth.2012.07.012) and the traditional spike features like trough-to-peak height and trough to peak delay.

288: Is there a reason why numbering jumps from 4 to 16 to 35 (25?)

363: Define what MaxRI and ExpectedRI are.

375: Define quantitatively what the homogeneity variable means, and what the completeness variable means. How is beta set?

388: Define s, d, maxR.

394: Define n,k.

401: Define a,b.

501: Why does Tied AE perform so poorly in the 3 external clustering measures? Did k-means randomly give a bad clustering?

It is common practice to highlight in bold the rows that perform the best in each column. Be consistent with the usage of commas in the CHS column.

Suggest some summary of the tables, perhaps using a rank aggregation method to give an overall ranking for each method (other methods are possible). Currently, many result values are presented but there are no overall statistics or significance testing (which may not be appropriate, but would be nice if this could be incorporated, say, by repeating simulations many times and analyzing the rankings of performance).

535: For real data, please provide plots of the spike waveforms after clustering (clusters color coded or in subplots) for visual evaluation.

599: While there are few parameters to tune (there are, in the learning algorithm for training the NN), the structure of the deep neural networks might be optimized per data set. Such tuning (and tuning of parameters for ICA and ISOMAP) may make a dramatic impact on the results. For maximum fairness, it would be best to at least do a cursory grid search over parameter values for the most sensitive parameters of each method; or show that the 6 metrics are not sensitive to parameter values (in supplementary). In anycase, all parameters used should be tabulated in the manuscript.

There is some randomness in the k-means clustering and NN training, so presenting the variability of the metrics in tables (in supplementary materials perhaps) would be useful.

6. PLOS authors have the option to publish the peer review history of their article (what does this mean?). If published, this will include your full peer review and any attached files.

Reviewer #1: **Yes: **Boreom Lee

Reviewer #2: No

---

## [Author Response · Author response to Decision Letter 0]

12 Jan 2023

Response to the viewers:

Reviewer #1

Reviewer #1 Point 1 - This research failed to identify the problem and purpose of the study; hence, its novelty was minimal. In addition, the manuscript was poorly organized, thus I would recommend that the authors revise and resubmit it. In the event of resubmission, the following are some of my suggestions: Please adhere to the correct template, conduct additional literature reviews, compare your results to earlier work, and use standard citation format. Hope for the best.

Reply: We have updated the manuscript structure such that it adheres to the submission guidelines, with modified section names and headings. With regard to the references, we have changed the Vancouver format to PLOS ONE’s version of Vancouver. Furthermore, we have restructured the paper to increase readability and clarity. We have added several explanations and literature citations to better illustrate the depth of our analysis. Moreover, we have extended the analysis of the methods from the 4 datasets described to all 95 synthetic datasets. The results are presented in the revised manuscript with the statistical evaluation using t-tests, with Bonferroni correction, shown in the Supplementary Material.

Reviewer #2

Reviewer #2 Point 1 - Overall the work is interesting and useful. The proposed methods show promise. The evaluation methods are fairly rigorous, but can be improved by looking at dependence on parameter tuning and computing statistics. The presentation needs to provide some missing details. It was fun to read a paper from Transylvania on Halloween.

Reply: First of all, we would like to thank the reviewer for the insightful comments. We struggled to address them as thorough as we could and performed a major revision of the manuscript. We have added the indicated parameter tuning of the classical feature extraction methods and have computed t-tests with a Bonferroni correction. These are now presented in the updated manuscript and the Supplementary Material document. 

Reviewer #2 Point 2 – Line 57: If the authors want to be more specific here, the problem isn't so much an imbalance, (a known imbalanced distribution can be handled) but unknown distribution, since changing firing rates mean different relative frequencies at different times.

Reply: We have added the suggested specifications in the updated manuscript. 

Reviewer #2 Point 3 – Line 59: Perhaps the authors can discuss the concept of single units and multiunits here and state what they mean when they write neurons in the manuscript.

Reply: We have added additional explanations on the concept of single and multi-units.

Reviewer #2 Point 3 – Line 61: Depending on the implementation, the 4th step is sometimes an offline (batch mode) clustering step, or alternatively an online classification step (using parameters previously found in offline batch mode clustering or by hand). This is a minor detail the authors can choose to include to be thorough.

Reply: We included a short description of the suggested information. 

Reviewer #2 Point 4 – Line 100: Around here would be a nice place for a brief discussion of why autoencoders are well suited for the spike sorting enterprise.

Reply: We added a short discussion on why autoencoders would be a good fit for spike sorting at the indicated location. 

Reviewer #2 Point 5 – Line 156: Consider mentioning wavelet transform (Yuan etal 10.1016/j.jneumeth.2012.07.012) and the traditional spike features like trough-to-peak height and trough to peak delay.

Reply: We have added the suggested information in the ‘State of the Art’ section of the updated manuscript. 

Reviewer #2 Point 6 – 288: Is there a reason why numbering jumps from 4 to 16 to 35 (25?)

Reply: The creators of these synthetic datasets created 95 of them. We have chosen four of these that we have found to be the most relevant based on the difficulties they provided. The number of clusters is incremented by four going from one dataset to another, such that it is becoming increasingly difficult to separate clusters as the overlap between them becomes larger. In the updated manuscript we now showcase an analysis over all 95 datasets and evaluated the significance of the results using t-tests with a Bonferroni correction. These can be found in Figure 5 of the updated manuscript and in figure 3 of the Supplementary Material.

Reviewer #2 Point 7 – Missing variable explanations

Line 363: Define what MaxRI and ExpectedRI are.

Line 375: Define quantitatively what the homogeneity variable means, and what the completeness variable means. How is beta set?

Line 388: Define s, d, maxR.

Line 394: Define n,k.

Line 401: Define a,b.

Reply: We have aggregated all these points under the same umbrella as they refer to the same type of overlook. In the updated manuscript we have added descriptions to every variable used in the equations. 

Reviewer #2 Point 8 – Line 501: Why does Tied AE perform so poorly in the 3 external clustering measures? Did k-means randomly give a bad clustering?

Reply: It is not a random bad clustering, as we show that the variability of K-Means is almost nonexistent in figure 1 of the Supplementary Material but rather a bad clustering due to the distribution of the datasets. The clusters, although separated, will be labelled incorrectly by K-Means due to their shape. We have added this information in the Discussion section of the updated manuscript.

Reviewer #2 Point 9 – It is common practice to highlight in bold the rows that perform the best in each column. Be consistent with the usage of commas in the CHS column.

Reply: We would like to thank the reviewer for highlighting this issue. We have amended the manuscript and corrected these inconsistencies.

Reviewer #2 Point 10 – Suggest some summary of the tables, perhaps using a rank aggregation method to give an overall ranking for each method (other methods are possible). Currently, many result values are presented but there are no overall statistics or significance testing (which may not be appropriate, but would be nice if this could be incorporated, say, by repeating simulations many times and analyzing the rankings of performance).

Reply: We have added the ranking of the Borda rank aggregation algorithm for the synthetic datasets (table 1) and the real data (table 3) in the Supplementary Material document. Additionally, as we evaluated the algorithms on all synthetic datasets in the revised manuscript, we have added the ranking on all 95 synthetic datasets (table 2) in the Supplementary Material as well.

Reviewer #2 Point 11 – Line 535: For real data, please provide plots of the spike waveforms after clustering (clusters color coded or in subplots) for visual evaluation.

Reply: We have added the requested evaluation in figure 8.

Reviewer #2 Point 12 – Line 599: While there are few parameters to tune (there are, in the learning algorithm for training the NN), the structure of the deep neural networks might be optimized per data set. Such tuning (and tuning of parameters for ICA and ISOMAP) may make a dramatic impact on the results. For maximum fairness, it would be best to at least do a cursory grid search over parameter values for the most sensitive parameters of each method; or show that the 6 metrics are not sensitive to parameter values (in supplementary). In anycase, all parameters used should be tabulated in the manuscript.

Reply: As the architecture of the autoencoders and their parameters are the same over all datasets, we have considered it fair to have the same type of evaluation for the other methods as well. We have implemented a grid search over all the parameters of ICA and Isomap and changed the results with the best results found and the parametrization used for each dataset. Because for Isomap the impact of parameters on the performance metrics numbers in the thousands, we have not added all the results in the manuscript but updated the best and mentioned the value in the caption of the table. We have added the information about the grid search and the evaluated parameters at the beginning of the ‘Performance Evaluation’ section.

Reviewer #2 Point 13 – There is some randomness in the k-means clustering and NN training, so presenting the variability of the metrics in tables (in supplementary materials perhaps) would be useful.

Reply: Indeed, we have added in the Supplementary Materials document figures that incorporate the variability of K-Means (figure 1) and the training of the autoencoder (figure 2).

Finally, we would like to thank the editor and the two reviewers for their insightful and helpful comments!

---

## [Decision Letter · Decision Letter 1]

5 Feb 2023

PONE-D-22-25738R1A Study of Autoencoders as a Feature Extraction Technique for Spike SortingPLOS ONE

Dear Dr. Ardelean,

Thank you for submitting your manuscript to PLOS ONE. After careful consideration, we feel that it has merit but does not fully meet PLOS ONE’s publication criteria as it currently stands. Therefore, we invite you to submit a revised version of the manuscript that addresses the points raised during the review process.

We look forward to receiving your revised manuscript.

Kind regards,

Yiming Tang, Ph.D.

Academic Editor

PLOS ONE

Reviewers' comments:

Reviewer's Responses to Questions

**Comments to the Author**

1. If the authors have adequately addressed your comments raised in a previous round of review and you feel that this manuscript is now acceptable for publication, you may indicate that here to bypass the “Comments to the Author” section, enter your conflict of interest statement in the “Confidential to Editor” section, and submit your "Accept" recommendation.

Reviewer #1: (No Response)

Reviewer #2: All comments have been addressed

2. Is the manuscript technically sound, and do the data support the conclusions?

Reviewer #1: Partly

Reviewer #2: Yes

3. Has the statistical analysis been performed appropriately and rigorously? 

Reviewer #1: Yes

Reviewer #2: Yes

4. Have the authors made all data underlying the findings in their manuscript fully available?

Reviewer #1: Yes

Reviewer #2: Yes

5. Is the manuscript presented in an intelligible fashion and written in standard English?

Reviewer #1: Yes

Reviewer #2: Yes

6. Review Comments to the Author

Reviewer #1: A Study of Autoencoders as a Feature Extraction Technique for Spike Sorting

- Eugen Richard Ardelean

Introduction:

The introduction should describe the overview of your research, background, or brief of existing research, which methods you approached, research problems, and statements.

To write an introduction you can follow the following steps:

Introduce your topic:

Describe the background:

Establish your research problem:

The goal of your research:

Map of your paper:

Line 113-117: The description in this paragraph does not correlate with the subsections presented in the article. I could not connect your statement to sections iii, iv, and v (section: v does not exist at all).

Materials And Methods:

Line 88 & 201: Figure 1 & 2 “Code” Most research is commonly used as Latent Space.

Method:

Line 247-253: It will be nice if you write steps as numbering instead of bullet points.

Discussion:

Line 531-534: Could you please explain fig 4, which one is under clustering and over clustering? Which color shows which information?

Line 533: What is sim14?

Line 542: How and why did you choose the number of epochs? Please provide the references.

Result:

Line 552, 622, 639: This description should be in the table title. Please make the sentences as the title and discuss the performance and analysis of the comparison.

In tables 3, 4, 5, 6, 7, 8, and so on you said comparison but did not explain the comparison. Why did you bold .99, .98, .27 33.910, and others? What was the exact result of your research comparison?

Figures 6, and 7 you marked as a,b,c,d… but did not find any discussion about it.

Please compare your results with those of previous studies. Or there was no previous work such as this one?

Conclusion:

“Performance by comparing them to state-of-the-art algorithms” no comparison with other studies has been presented. Although, a comparison was made with conventional spike sorting techniques. This does not necessarily translate to state-of-the-art.

Final Comment:

Please accept my gratitude for the outstanding work you have done. More research on the topic at hand is strongly suggested. Your thesis should be written as a scholarly manuscript. In contrast to a research article, a thesis is written in a different format. I will suggest taking a look at the articles that were submitted before in the journal and the way the author organized their good work. Regarding future projects, you should conduct further literature reviews. We hope that your future writing ventures are as successful as your past ones have been.

Reviewer #2: The authors have addressed all of my concerns and the manuscript is much improved.

The authors may wish to pay attention to legibility of words in figures during proof, as the words were very small in the review pdf.

7. PLOS authors have the option to publish the peer review history of their article (what does this mean?). If published, this will include your full peer review and any attached files.

Reviewer #1: No

Reviewer #2: No

---

## [Author Response · Author response to Decision Letter 1]

8 Feb 2023

Dear Editor(s),

We thank the reviewers for taking the time to read our manuscript and present their concerns. We have thoroughly revised the manuscript and addressed the points highlighted by the reviewers. We hope that the manuscript is suitable for publication now.

Response to reviewers:

Reviewer #1

Reviewer #1 Point 1 – Introduction

The introduction should describe the overview of your research, background, or brief of existing research, which methods you approached, research problems, and statements.

To write an introduction you can follow the following steps:

Introduce your topic:

Describe the background:

Establish your research problem:

The goal of your research:

Map of your paper:

Reply: We have taken into account the points of all reviewers and we have now reformatted the introduction according to their suggestion, to the best of our ability. It is now separated into three subsections. Section 1.1 introduces the topic of spike sorting—the main topic of the paper. Section 1.2 cursory describes autoencoders and outlines the necessary background. Finally, section 1.3 establishes the challenges of spike sorting that define our research problem and states the goal of the research. As suggested by the reviewer, the introduction ends with a map of the paper. 

Reviewer #1 Point 2 – Line 113-117: The description in this paragraph does not correlate with the subsections presented in the article. I could not connect your statement to sections iii, iv, and v (section: v does not exist at all). 

Reply: A major restructuring of the manuscript has been made in a previous stage of revision and we overlooked modifying this paragraph. We would like to thank the reviewers for point it out for us. We have now fixed this.

Reviewer #1 Point 3 – Materials And Methods:

Line 88 & 201: Figure 1 & 2 “Code” Most research is commonly used as Latent Space.

Reply: The figures have been modified such that instead of ‘Code’, they now contain ‘Latent space’.

Reviewer #1 Point 3 – Method:

Line 247-253: It will be nice if you write steps as numbering instead of bullet points.

Reply: The steps of pretraining have been modified from bullet points to numbering.

Reviewer #1 Point 4 – Discussion:

Line 531-534: Could you please explain fig 4, which one is under clustering and over clustering? Which color shows which information?

Reply: Indeed, we have not explained this properly. In case of fig 4, neither of them shows either under or overclustering, because the colors display the ground truth and not the result of a clustering algorithm. Under and overclustering are generated by clustering algorithms in specific cases, when compared to a ground truth. When two (or more) clusters in the ground truth are labelled as one by the clustering algorithm, it is considered underclustering. While, when a single cluster in the ground truth is considered as two (or more) clusters by the clustering algorithm, we can say that the algorithm is overclustering. Additional explanations have now been added to the manuscript.

Reviewer #1 Point 5 – Line 533: What is sim14?

Reply: Simulation 14 is another dataset that is encompassed in the 95 synthetic datasets used [45]. We have chosen to show the impact of alignment on simulation 14 because it has a lower number of clusters and samples and is therefore easier to visualize. We have added additional information about it, where we also discuss the other simulations, in section 2.3.1.

Reviewer #1 Point 6 – Line 542: How and why did you choose the number of epochs? Please provide the references.

Reply: In table 2, we show the impact of the number of epochs (through a selected number of samples) on the performance of the model with regard to the metrics. We have chosen the number of epochs based on the results of the metrics and the loss of the model. We add below a plot of the loss values across epochs, showing that at around 50 epochs the loss of model saturates and increasing the number of epochs would not bring significant benefit. This figure has also been added to the Supporting Information and is now referenced in the manuscript.

Reviewer #1 Point 7 – Result:

Line 552, 622, 639: This description should be in the table title. Please make the sentences as the title and discuss the performance and analysis of the comparison.

Reply: The indicated lines have been modified (and explanations have been added) according to the suggestions of the reviewer.

Reviewer #1 Point 8 – In tables 3, 4, 5, 6, 7, 8, and so on you said comparison but did not explain the comparison. Why did you bold .99, .98, .27 33.910, and others? What was the exact result of your research comparison?

Reply: The caption of each table has been modified according to the suggestion at the previous point. The bolded values highlight the best values for each metric (we say best instead of highest because the DBS metric is inverse, with lower values indicating better clustering), thus indicating the best performing methods according to each metric. We now mention this in the caption, for each table. The overall conclusions that can be extracted from the analysis are now discussed in section 4. 

Reviewer #1 Point 9 – Figures 6, and 7 you marked as a,b,c,d… but did not find any discussion about it.

Reply: The reviewer is right, this was a mistake. Figures 6 and 7 now no longer contain any alphabetical labelling, as this is not necessary and not used. 

Reviewer #1 Point 10 – Please compare your results with those of previous studies. Or there was no previous work such as this one?

Reply: We have referenced two recent papers that study autoencoders in spike sorting ([15] and [16]). We have now described their results in the introduction. However, one major issue preventing a direct comparison to our approach is that these previous studies have been limited to synthetic datasets that contain a low number of clusters and, in addition, their evaluation was restricted to accuracy for [15] and, accuracy and silhouette score for [16]. We already discussed that accuracy is not a suitable metric for the problem of spike sorting because: i) it can only be used for synthetic datasets that contain a ground truth; and ii) for datasets that have a low amount of imbalance. Imbalance is an inherent problem in spike sorting because different neurons usually have very different firing rates, thus generating clusters of different densities. The typical imbalance observed in spike datasets renders accuracy quite problematic as a performance metric. We now elaborate on these concerns when discussing performance metrics in the manuscript. In [16], the authors propose contractive autoencoders for spike sorting, which is one of the variants that we also evaluate. Thus, we now mention the previous studies we could find, we discuss their results, but also point out the limitations in relating to their results due to the use of accuracy as a performance metric on imbalanced data.

Reviewer #1 Point 11 – Conclusion:

“Performance by comparing them to state-of-the-art algorithms” no comparison with other studies has been presented. Although, a comparison was made with conventional spike sorting techniques. This does not necessarily translate to state-of-the-art.

Reply: We have now reformulated this part of the conclusions. 

Reviewer #2

The authors have addressed all of my concerns and the manuscript is much improved.

The authors may wish to pay attention to legibility of words in figures during proof, as the words were very small in the review pdf.

Reply: We would like to thank the reviewer for the insightful comments in the previous stage of revision.

Finally, we would like to thank the editor and the two reviewers for their insightful and helpful comments!

---

## [Decision Letter · Decision Letter 2]

23 Feb 2023

A Study of Autoencoders as a Feature Extraction Technique for Spike Sorting

PONE-D-22-25738R2

Dear Dr. Ardelean,

We’re pleased to inform you that your manuscript has been judged scientifically suitable for publication and will be formally accepted for publication once it meets all outstanding technical requirements.

Kind regards,

Yiming Tang, Ph.D.

Academic Editor

PLOS ONE

Additional Editor Comments (optional):

Reviewers' comments:

Reviewer's Responses to Questions

**Comments to the Author**

1. If the authors have adequately addressed your comments raised in a previous round of review and you feel that this manuscript is now acceptable for publication, you may indicate that here to bypass the “Comments to the Author” section, enter your conflict of interest statement in the “Confidential to Editor” section, and submit your "Accept" recommendation.

Reviewer #1: All comments have been addressed

Reviewer #2: All comments have been addressed

2. Is the manuscript technically sound, and do the data support the conclusions?

Reviewer #1: Yes

Reviewer #2: Yes

3. Has the statistical analysis been performed appropriately and rigorously? 

Reviewer #1: Yes

Reviewer #2: Yes

4. Have the authors made all data underlying the findings in their manuscript fully available?

Reviewer #1: Yes

Reviewer #2: Yes

5. Is the manuscript presented in an intelligible fashion and written in standard English?

Reviewer #1: Yes

Reviewer #2: Yes

6. Review Comments to the Author

Reviewer #1: All of my concerns have been resolved by the authors, and the manuscript has greatly improved. Thank you to the author for your significant improvement. I wish you the best of success in your future endeavors. Again, I recommend that you conduct a more extensive literature review for your work and follow the guidelines provided by the publisher.

Reviewer #2: The authors have previously addressed my concerns. The revisions in response to the other reviewer have further improved the manuscript. I recommend acceptance.

7. PLOS authors have the option to publish the peer review history of their article (what does this mean?). If published, this will include your full peer review and any attached files.

Reviewer #1: No

Reviewer #2: No

---

## [Editor Report · Acceptance letter]

28 Feb 2023

PONE-D-22-25738R2 

A Study of Autoencoders as a Feature Extraction Technique for Spike Sorting 

Dear Dr. Ardelean:

I'm pleased to inform you that your manuscript has been deemed suitable for publication in PLOS ONE. Congratulations! Your manuscript is now with our production department. 

Kind regards, 

on behalf of

Professor Yiming Tang 

Academic Editor

PLOS ONE